# Bayesian inference of population prevalence

**Robin AA Ince[1]\*, Angus T Paton[1], Jim W Kay[2], Philippe G Schyns[1]**

[1]School of Psychology and Neuroscience, University of Glasgow, Glasgow, United Kingdom; [2]Department of Statistics, University of Glasgow, Glasgow, United Kingdom

**Abstract** Within neuroscience, psychology, and neuroimaging, the most frequently used statistical approach is null hypothesis significance testing (NHST) of the population mean. An alternative approach is to perform NHST within individual participants and then infer, from the proportion of participants showing an effect, the prevalence of that effect in the population. We propose a novel Bayesian method to estimate such population prevalence that offers several advantages over population mean NHST. This method provides a population-level inference that is currently missing from study designs with small participant numbers, such as in traditional psychophysics and in precision imaging. Bayesian prevalence delivers a quantitative population estimate with associated uncertainty instead of reducing an experiment to a binary inference. Bayesian prevalence is widely applicable to a broad range of studies in neuroscience, psychology, and neuroimaging. Its emphasis on detecting effects within individual participants can also help address replicability issues in these fields.

## Introduction

Within neuroscience, psychology, and neuroimaging, the common experimental paradigm is to run an experiment on a sample of participants and then infer and quantify any effect of the experimental manipulation in the population from which the sample was drawn. For example, in a psychology experiment, a particular set of stimuli (e.g. visual or auditory stimuli) might be presented to a sample of human participants, who are asked to categorise the stimuli or perform some other task. Each participant repeats the procedure several times with different stimuli (experimental trials), and their responses and reaction times are recorded. In a neuroimaging experiment, the same procedure is employed with neuroimaging signals recorded in addition to behavioural responses. The researcher analyses these responses to infer something about brain function in the population from which the participants were drawn.

In this standard experimental paradigm, the implicit goal is usually to determine the presence of a causal relationship between the experimental manipulation and the response of interest. For example, between a stimulus property and the neural activity in a particular brain region, or between neural activity and a behavioural measure (e.g. accuracy, reaction time). A properly controlled experimental design in which other extraneous factors are held constant (i.e. a randomised control trial) enables a causal interpretation of a correlational relationship (*Angrist and Pischke, 2014*; *Pearl, 2009*). We use statistical tools to evaluate the measured effect and ensure that we are not being fooled by randomness—that is, it should be unlikely our observed result was produced by random fluctuations with no effect present. This is often formalised as null hypothesis significance testing (NHST). We reject the null hypothesis (often that the population mean is zero) when the probability of observing a given effect size (or larger) is less than some prespecified value (often $p$=0.05) if the null hypothesis was true (i.e. if the population mean really was zero). Simply stated, we would be unlikely to obtain the observed effect size if the null hypothesis was true.

Researchers performing such studies usually wish to infer something about the population from which the experimental participants are selected (*Holmes and Friston, 1998*), rather than about the specific sample of participants that were examined (as in a case study). Importantly, any statistical

**\*For correspondence:**
robin.ince@glasgow.ac.uk

**Competing interest:** The authors declare that no competing interests exist.

**eLife digest** Scientists use statistical tools to evaluate observations or measurements from carefully designed experiments. In psychology and neuroscience, these experiments involve studying a randomly selected group of people, looking for patterns in their behaviour or brain activity, to infer things about the population at large.

The usual method for evaluating the results of these experiments is to carry out null hypothesis statistical testing (NHST) on the population mean – that is, the average effect in the population that the study participants were selected from. The test asks whether the observed results in the group studied differ from what might be expected if the average effect in the population was zero. However, in psychology and neuroscience studies, people's brain activity and performance on cognitive tasks can differ a lot. This means important effects in individuals can be lost in the overall population average.

Ince et al. propose that this shortcoming of NHST can be overcome by shifting the statistical analysis away from the population mean, and instead focusing on effects in individual participants. This led them to create a new statistical approach named Bayesian prevalence. The method looks at effects within each individual in the study and asks how likely it would be to see the same result if the experiment was repeated with a new person chosen from the wider population at random.

Using this approach, it is possible to quantify how typical or uncommon an observed effect is in the population, and the uncertainty around this estimate. This differs from NHST which only provides a binary 'yes or no' answer to the question, 'does this experiment provide sufficient evidence that the average effect in the population is not zero?' Another benefit of Bayesian prevalence is that it can be applied to studies with small numbers of participants which cannot be analysed using other statistical methods.

Ince et al. show that the Bayesian prevalence can be applied to a range of psychology and neuroimaging experiments, from brain imaging to electrophysiology studies. Using this alternative statistical method could help address issues of replication in these fields where NHST results are sometimes not the same when studies are repeated.

inference from a sample to the population requires a model of the population itself. The ubiquitous approach used in psychology and neuroimaging is to model the effect in the population with a normal distribution and perform inference on the mean of this model: the population mean (see Materials and methods). For example, the null hypothesis is often that the population mean is zero, and the probability of the observed sample data is computed under this assumption, taking into account the variance between individuals in the sample of participants.

However, an alternative and equally valid question is to ask how typical is an effect in the population (*Friston et al., 1999b*). In this approach, we infer an effect in each individual of the sample, and from that infer the *prevalence* of the effect in the population—that is, the proportion of the population that would show the effect, if tested (*Allefeld et al., 2016*; *Donhauser et al., 2018*; *Friston et al., 1999a*; *Rosenblatt et al., 2014*). The results obtained using the population mean (vs. population prevalence) can differ, particularly when effects are heterogenous across participants.

Here, we argue that in many experimental applications in psychology and neuroscience, the individual participant is the natural replication unit of interest (*Little and Smith, 2018*; *Nachev et al., 2018*; *Smith and Little, 2018*; *Thiebaut de Schotten and Shallice, 2017*). This is because many aspects of cognition, and the neural mechanisms underlying them, are likely to vary across individuals. Therefore, we should seek to quantify effects and ensure that our results can be reliably distinguished from chance within individual participants. We argue that with such a shift in perspective towards experimental assessment within individual participants, we should also shift our statistical focus at the population level from NHST of the population mean to estimating the population prevalence: the proportion of individuals in the population who would show a true positive above-chance effect in a specific experiment. This can be thought of as the expected within-participant replication probability. Although we focus here on a broad class of experiments in psychology and neuroimaging that feature human participants and non-invasive recording modalities, the arguments we present are general and apply equally well to other experimental model organisms or sampling units.

To support this shift in perspective, we present a simple Bayesian method to estimate population prevalence based on the results of within-participant NHST, including prevalence differences between groups of participants or between tests on the same participants. This approach can also be applied without explicit within-participant NHST to estimate prevalence of different effect sizes, giving a new view on what can be learned about a population from an experimental sample. We suggest that applying Bayesian population prevalence in studies that are sufficiently powered within individual participants could address many of the recent issues raised about replicability in psychology and neuroimaging research (*Benjamin et al., 2018*; *Ioannidis, 2005*). Bayesian prevalence provides a population prevalence estimate with associated uncertainty and therefore avoids reducing an entire experiment to a binary NHST inference on a population mean (*McShane et al., 2019*).

## Results

### Population vs. within-participant statistical tests

To illustrate, we simulate data from the standard hierarchical population model that underlies inference of a population mean effect based on the normal distribution (*Friston, 2007*; *Holmes and Friston, 1998*; *Penny and Holmes, 2007*) (see Materials and methods).

We emphasise that this standard hierarchical population simulation is the simplest possible illustrative example, intended to demonstrate the different perspective of within-participant vs. population mean inference. The model simulated here is used implicitly in almost every random-effects inference employed in psychology and neuroimaging studies, in which the tested null hypothesis is that the population distribution is normal with zero mean and non-zero variance. We also emphasise that the simulated data values within individuals are considered to represent the *effect* of a controlled experimental manipulation—that is, a causal relationship. However, this could be any causal relationship that we can quantify in psychology, neuroscience, or neuroimaging. For example, we could consider a within-participant difference in reaction time between two classes of stimuli, a coefficient or contrast from a linear model (e.g. a General Linear Model of fMRI data), a cross-validated out-of-sample predictive correlation from a high-dimensional stimulus encoding model (e.g. a model predicting auditory cortex MEG responses to continuous speech stimuli), a rank correlation of dissimilarity matrices in a Representational Similarity Analysis, or parameters of computational models of decision making (e.g. the Diffusion Drift Model) or learning. When evaluating the results of the simulations of *Figure 1*, it is important to keep in mind that these results are meant to represent any statistical quantification of any experimental manipulation.

*Figure 1* illustrates the results of four simulations that differ in the true population mean, μ (A, B: $\mu = 0$; C, D, $\mu = 1$) and in the number of trials performed per participant (A, C: 20 trials; B, D: 500 trials). For each simulation, we performed inference based on a standard two-sided, one-sample t-test against zero at two levels. First, we applied the standard summary statistic approach: we took the mean across trials for each participant and performed the second-level population t-test on these per-participant mean values. This provides an inference on the population mean, taking into account between-participant variation. This is equivalent to inference on the full hierarchical model in a design where participants are separable (*Holmes and Friston, 1998*; *Penny and Holmes, 2007*; *Penny et al., 2003*) (see Materials and methods). The modelled population distribution is plotted as a solid curve, coloured according to the result of the population mean inference (orange for significant population mean, blue for non-significant population mean). Second, we performed inference within each participant, applying the t-test on within-participant samples, separately for each participant. The sample mean ± s.e.m. is plotted for each participant (orange for significant participants, blue for non-significant participants).

The population mean inference correctly fails to reject the null hypothesis for *Figure 1* panels A and B (ground truth $\mu = 0$) and correctly rejects the null in panels C and D (ground truth $\mu = 1$). But consider carefully *Figure 1B,C*. With 500 trials in panel B, 32/50 participants (orange markers) show a significant within-participant result. The probability of this happening by chance, if there was no effect in any members of the population, can be calculated from the cumulative density function of the binomial distribution. In this case, it is tiny—for a test with false positive rate $\alpha = 0.05$, and no effect in any individual, $p < 2.2 \times 10^{-16}$ (below 64-bit floating-point precision). Compare that to $p = 0.008$ for the population t-test in panel C. Thus, the panel B results provide

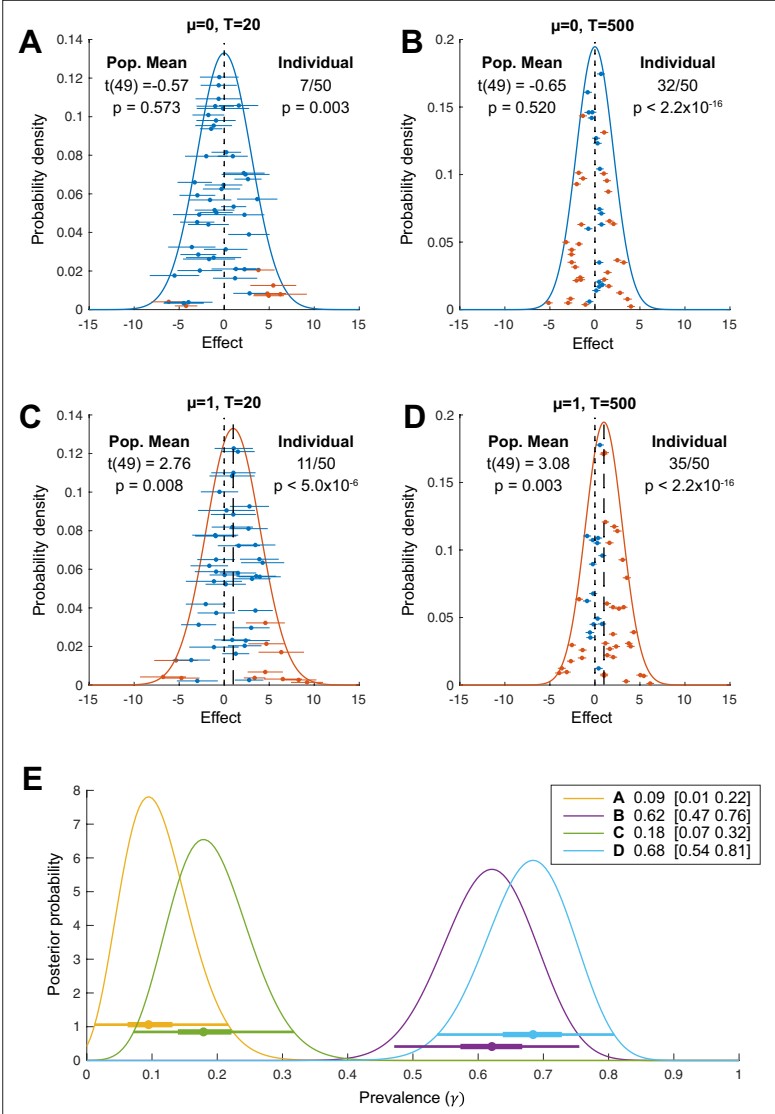

**Figure 1.** Population vs individual inference. For each simulation, we sample $N = 50$ individual participant mean effects from a normal distribution with population mean $\mu$ (**A, B**) $\mu = 0$; (**C, D**) $\mu = 1$ and between-participant standard deviation $\sigma_b = 2$. Within each participant, $T$ trials (**A, C**) $T = 20$; (**B, D**) $T = 500$ are drawn from a normal distribution with the participant-specific mean and a common within-participant standard deviation $\sigma_w = 10$ (*Baker et al., 2020*). Orange and blue indicate, respectively, exceeding or not exceeding a $p=0.05$ threshold for a t-test at the population level (on the within-participant means, population normal density curves) or at the individual participant level (individual sample means ± s.e.m.). (**E**): Bayesian posterior distributions of population prevalence of true positive results for the four simulated data sets (**A–D**). Circles show Bayesian maximum a posteriori (MAP) estimates. Thick and thin horizontal lines indicate 50% and 96% highest posterior density intervals (HPDIs), respectively. MAP (96% HPDI) values are shown in the legend.

much stronger statistical evidence of a non-zero effect *at the population level:* the observed results are very unlikely if no individuals in the population have a non-zero effect in this test. This statistical evidence would be ignored in analyses based only on the population mean. Considering inference at the individual level, panel C results (11/50 significant) have $p=4.9\times10^{-6}$ if there was no effect within any individuals (i.e. the proportion of the population showing an effect was zero). Thus, even panel C, which simulates an experiment with only 20 trials per participant, provides stronger evidence for a population effect from the within-participant perspective than from the population mean perspective.

Obviously, these two different p-values are calculated from two different null hypotheses. The population t-test tests the null hypothesis that the population mean is zero, assuming that individual means follow a normal distribution:

$$H_0: \ \mu_{pop} = 0$$

while the p-value for the number of significant within-participant tests comes from the null hypothesis that there is no effect in any individual in the population, termed the *global null* (***Allefeld et al., 2016***; ***Donhauser et al., 2018***; ***Nichols et al., 2005***):

$$H_0: \ \mu_i = 0 \ for \ all \ i$$

where i indexes all members of the population. These are testing different questions, but importantly both are framed at the population level and both provide a population-level inference. Clearly, the global null itself is quite a blunt tool. The goal of our Bayesian prevalence method is to quantify within-participant effects at the population level in a more meaningful and graded way. We agree that it is important to align 'the meaning of the quantitative inference with the meaning of the qualitative hypothesis we're interested in evaluating' (***Yarkoni, 2020***). Often, when the goal of the analysis is to infer the presence of a causal relationship within individuals in the population, the within-participant perspective may be more appropriate. We argue that performing NHST at the individual participant level is preferable for conceptual reasons in psychology and neuroimaging, but also for practical reasons related to the replicability crisis (see Discussion).

Although we have so far considered only the global null, the simulations show that the within-participant perspective can give a very different impression of the evidence for a population-level effect from a data set. The results in ***Figure 1B*** represent strong evidence of a non-zero effect within many participants, albeit one that is approximately balanced across participants between positive and negative directions. We do not intend to suggest that such two-sided effects are particularly common but would highlight how they are implicitly assumed in traditional population mean inference models. ***Figure 2*** provides further examples of how the within-participant and population mean approaches can diverge in situations with effects in one direction. In fact, if the researcher believes that all effects

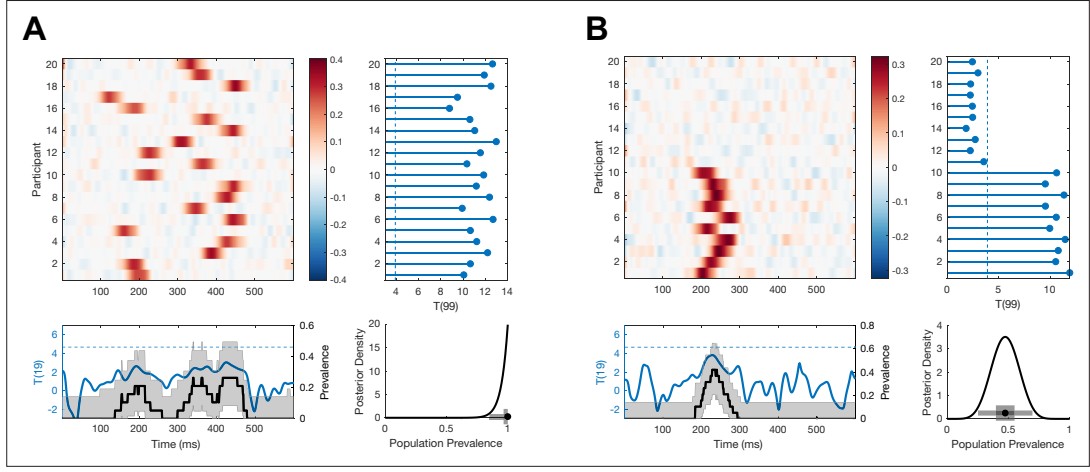

**Figure 2.** Simulated examples where Bayesian prevalence and second-level t-tests diverge. EEG traces are simulated for 100 trials from 20 participants as white noise, $N(0, 1)$, with an additive Gaussian activation ($\sigma$ = 20 ms) with amplitudes drawn from a uniform distribution on [0 0.6]. For each simulation, mean traces are shown per participant (upper-left panel). A second-level t-test is performed at each time point separately (blue curve, lower-left panel), dashed line shows the *p*=0.05 threshold, Bonferroni corrected over time points. A within-participant t-test is performed at each time point and for each participant separately (right-hand panel); the blue points show the maximum T-statistic over time for each participant, and the dashed line shows the *p*=0.05 Bonferroni corrected threshold. Lower-right panel shows posterior distribution of population prevalence for an effect in the analysis window. Black curves (lower-left panel) show the prevalence posterior at each time point (black line maximum a posteriori [MAP], shaded area 96% highest posterior density interval [HPDI]). (**A**) An effect is simulated in all participants, with a peak time uniformly distributed in the range 100–400 ms. (**B**) An effect is simulated in 10 participants, with a peak time uniformly distributed in the range 200–275 ms.

in an experiment are positive, then the standard null model illustrated here is inappropriate. The traditional population mean t-test is actually providing a fixed-effects analysis based on the global null rather than the random-effects interpretation of generalisation to the population that is normally applied (*Allefeld et al., 2016*).

## Estimating population prevalence

The p-values under the global null are obtained from the cumulative density function of the binomial distribution, based on a within-participant false positive rate $\alpha = 0.05$. However, we can also model the number of above-threshold individuals in a sample when the population prevalence of true positive (significant) test results is $\gamma$ (see Materials and methods). Consider a within-participant test with a false positive rate $\alpha$. In this case, the distribution of the number of significant test results follows a binomial distribution with success probability $\theta = (1 - \gamma) \alpha + \gamma$. Here, we present a Bayesian approach to estimate the population prevalence of true positive test results, $\gamma$, from this binomial model of within-participant testing. Alternatively, frequentist inference approaches can be used (*Allefeld et al., 2016*; *Donhauser et al., 2018*; *Friston et al., 1999a*). Note that we are not estimating the population prevalence of the ground truth binary status of the tested effect. We could only obtain a lower bound on this quantity because there might be individuals with an effect too small to detect with our test. Therefore, here we focus throughout on the prevalence of true positive test results— that is, the proportion of the population we would expect to give a true positive test result given the specific experimental test procedure considered.

The Bayesian approach provides a full posterior distribution for $\gamma$, from which we can obtain the maximum a posteriori (MAP) estimate, together with measures of uncertainty—for example, highest posterior density intervals (HPDIs) or lower bound posterior quantiles. *Figure 1E* shows the Bayesian posteriors, MAPs, and HPDIs for the four simulated data sets in *Figure 1A–D*. Even though there is no population mean effect in *Figure 1B*, the MAP estimate of the prevalence is 0.62 (96% HPDI: [0.47 0.76]). Given the data, the probability that the population prevalence is greater than 47% is higher than 0.96. Based on this result, we would consider it highly likely that more than 47% of the population would show a true positive effect if tested in the same experiment with 500 trials.

*Figure 2* demonstrates two situations where Bayesian prevalence gives a different statistical assessment of the presence of an effect in the population compared to a second-level t-test for a non-zero population mean. We simulated a simple EEG experiment with 100 trials repeated in each of 20 participants. Template Gaussian event-related potentials (ERPs) are added on each trial with fixed width and uniformly distributed amplitude, with a specific peak time per participant. Both within-participant and second-level t-tests are Bonferroni corrected over the 600 time points considered. *Figure 2A* shows a simulation where all participants have an effect, with peak times drawn from a uniform distribution over the 100–400 ms range. There is strong evidence for an effect in each participant, and the estimated prevalence is high: 1 [0.85 1] (MAP [96% HPDI]). However, because the effects are not aligned in time across participants, there are no time points at which the null hypothesis of zero population mean can be rejected. In the simulation shown in *Figure 2B*, only 10/20 participants demonstrate an effect, with peak times drawn from a uniform distribution over the range 200–275 ms. Here, effects can be reliably detected in those 10 participants, leading to an estimated prevalence of 0.47 [0.25 0.70]. However, because the effects are not present in all participants, there are no time points when the null hypothesis of zero population mean can be rejected. Interestingly, plotting the prevalence posterior distribution as a function of time does reveal evidence for an effect in the population during the time window of the simulated effect.

## Estimating differences in prevalence

Often the scientific question of interest might involve comparing an effect between groups of participants or between different experimental conditions in the same set of participants. In the first case, a researcher might, for example, be interested in examining differences in a behavioural or neuroimaging effect between patients and healthy controls, or between participants from two different cultures. In the second case, a researcher might be interested in determining the effect of an intervention on a particular measured effect, testing the same set of participants on two occasions, once with an intervention and once in a control condition. From the population mean perspective, these questions would typically be addressed with a two-sample unpaired t-test for the first case and a paired

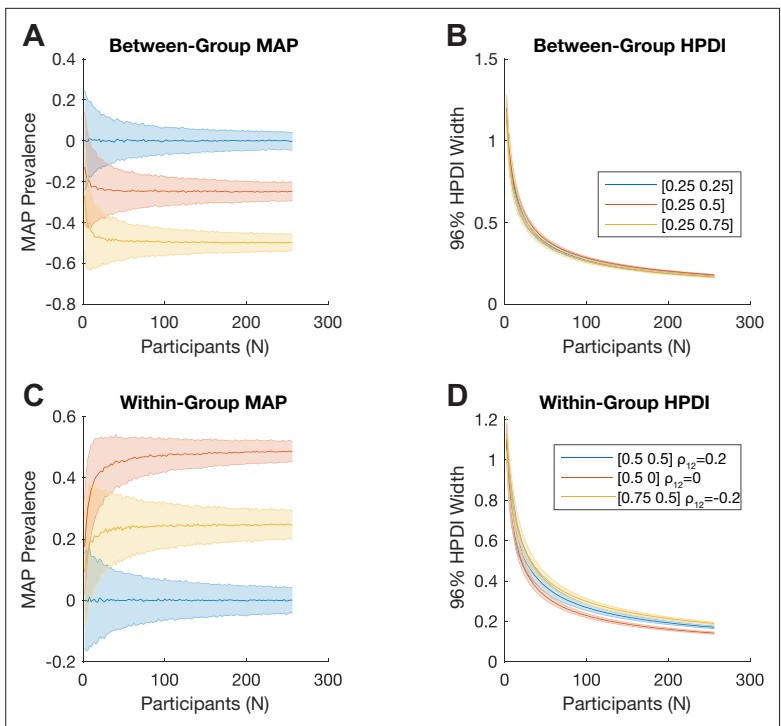

**Figure 3.** Bayesian inference of difference of prevalence. (**A, B**) We consider two independent groups of participants with population prevalence of true positives $[\gamma_1, \gamma_2]$ of [25%, 25%] (blue), [25%, 50%] (red), and [25%, 75%] (yellow). We show how (**A**) the Bayesian MAP estimate and (**B**) 96% highest posterior density interval (HPDI) width of the estimated between-group prevalence difference $\gamma_1 - \gamma_2$ scale with the number of participants. (**C, D**) We consider two tests applied to the same sample of participants. Here, each simulation is parameterised by the population prevalence of true positives for the two tests, $[\gamma_1, \gamma_2]$, as well as $\rho_{12}$, the correlation between the (binary) test results across the population. We show this for [50%, 50%] with $\rho_{12} = 0.2$ (blue), [50%, 0%] with $\rho_{12} = 0$ (red), and [75%, 50%] with $\rho_{12} = -0.2$ (yellow). We show how (**C**) the Bayesian maximum a posteriori (MAP) estimate and (**D**) 96% HPDI width of the estimated within-group prevalence difference $\gamma_1 - \gamma_2$ scale with the number of participants.

t-test for the second. From the prevalence perspective, the question would be whether the prevalence of true positive results differs between two sampled populations (in the first case) or between two experimental tests (in the second case). We therefore provide additional functions (see Materials and methods) to directly estimate the difference in prevalence of true positive test results for these two comparisons, which we term between-group (same test applied to two different samples) and within-group (two different tests applied to a single sample).

To estimate the difference in prevalence of true positive results for a given test between two samples from separate populations (e.g. patients vs. healthy controls), the input data required is the count of positive results and the total number of participants in each group. We illustrate this with a simulated data set. We specify the true prevalence in the two populations as $\gamma_1 = 0.75$ and $\gamma_2 = 0.25$, respectively, and draw a random sample based on the respective binomial distributions with parameters $\theta_i$ (see Materials and methods). We simulate $N_1 = 60$ participants in the first group and $N_2 = 40$ participants in the second group. The results of one such draw give $k_1 = 45$, $k_2 = 11$ positive tests in each group, respectively. In this case, the MAP [96% HPDI] prevalence difference $\gamma_1 - \gamma_2$, calculated from these four values $(k_1, k_2, N_1, N_2)$, is 0.49 [0.29 0.67], which closely matches the ground truth (0.5). *Figure 3A and B* shows how the between-group posterior prevalence difference estimates scale with the number of participants for three different simulations.

To estimate the difference in prevalence of true positive results between two different tests applied to the same sample of participants, the input parameters are the number of participants significant in both tests, the number significant only in each of the two tests, and the total number of participants. We simulate two tests applied to a group of $N = 50$ participants. Each test detects a certain property

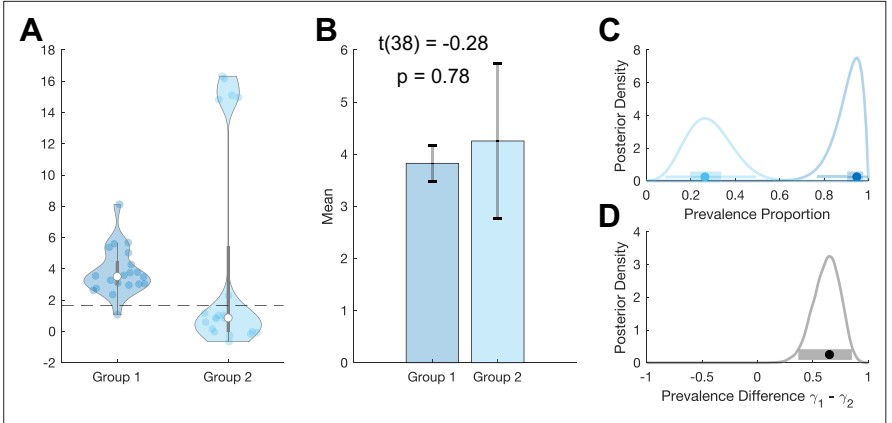

**Figure 4.** Example where between-group prevalence diverges from two-sample t-test. We simulate standard hierarchical Gaussian data for two groups of 20 participants, $T = 100$, $\sigma_w = 10$, $N = 20$ per group. (**A**) Group 1 participants are drawn from a single population Gaussian distribution with $\mu = 4$, $\sigma_b = 1$. Group 2 participants are drawn from two Gaussian distributions. 75% of participants are drawn from $N(0, 0.01)$ and 25% of participants are drawn from $N(16, 0.5)$. Dashed line shows the p=0.05 within-participant threshold (one-sample t-test). The means of these two groups are not significantly different (**B**), but they have very different prevalence posteriors (**C**). The posterior distribution for the difference in prevalence shows the higher prevalence in group 1: 0.61 [0.36 0.85] (MAP [96% HPDI]) (**D**). MAP: maximum a posteriori; HDPI: highest posterior density interval.

with false positive rate $\alpha = 0.05$. The ground truth prevalences of true positives for the two tests are $\gamma_1 = 0.5$ and $\gamma_2 = 0.25$, respectively, and the correlation between positive results is $\rho_{12} = 0.2$ (i.e. participants who test positive on one test are more likely to also test positive on the other test). The results of one random draw from this model give: (see Materials and methods) $k_{11} = 8$ participants with a significant result in both tests; $k_{10} = 19$ participants with a significant result in the first test but not the second; and $k_{01} = 5$ participants with a significant result in the second but not the first. In this case, the MAP [96% HPDI] prevalence difference $\gamma_1 - \gamma_2$, calculated from these four values ($k_{11}, k_{10}, k_{01}, N$), is 0.28 [0.08 0.46], which again matches the ground truth (0.25). *Figure 3C,D* shows how the within-group posterior prevalence difference estimates scale with the number of participants for three different ground truth situations, given as $\begin{bmatrix} \gamma_1\ \gamma_2 \end{bmatrix} \rho_{12}$.

Both these approaches are implemented using Monte Carlo methods, and the functions return posterior samples (*Gelman, 2014*), which can be used to calculate other quantities, such as the posterior probability that one test or group has a higher prevalence than the other. The posterior log odds in favour of this hypothesis can be computed by applying the logit function to the proportion of posterior samples in favour of a hypothesis. In the between-group example above, the proportion of posterior samples in favor of the hypothesis $\gamma_1 > \gamma_2$ is 0.9999987, corresponding to log posterior odds of 13.5. In the above within-group comparison, the proportion of posterior samples in favour of the hypothesis $\gamma_1 > \gamma_2$ is 0.9979451, corresponding to log posterior odds of 6.2 (each computed from 10 million posterior samples).

The additional example in *Figure 4* demonstrates how between-group prevalence differences can occur between two populations with the same mean. We simulated two groups of participants where group 1 was from a homogenous population with a non-zero mean whereas group 2 comprised participants drawn from two different distributions, one with zero mean and one with a large mean. The two samples (and populations) have similar means, with no significant difference between them (two-sample t-test p=0.78). However, considering the prevalence in the two populations clearly reveals their difference (*Figure 4C*), which is formalised with the posterior distribution of the prevalence difference between the two populations (*Figure 4D*).

## Prevalence as a function of effect size

In the above, we focused on performing explicit statistical inference within each participant. A possible criticism of this approach is that the within-participant binarisation of a continuous effect size can lose information. If the null distribution is the same for each participant, then the within-participant inference involves comparing each participant's effect size, $E_p$, to a common statistical threshold $\hat{E}$. The prevalence

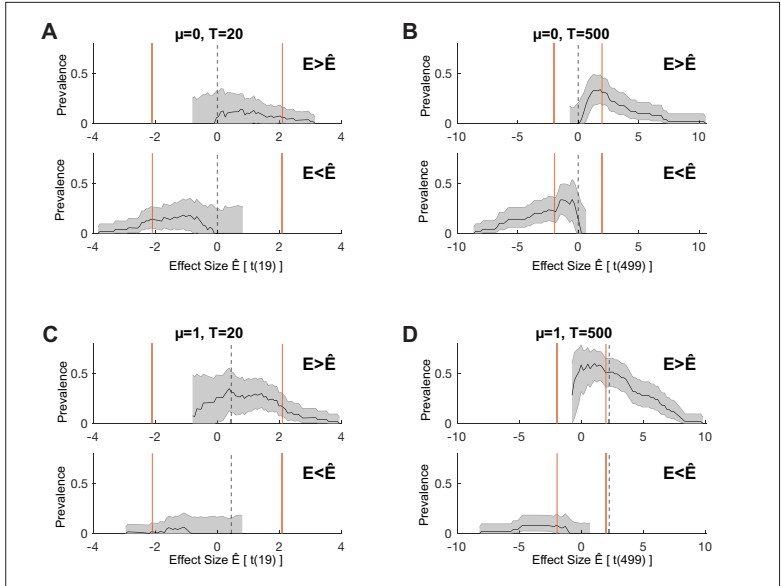

**Figure 5.** One-sided prevalence as a function of effect size. We consider the same simulated systems shown in *Figure 1*, showing both right-tailed ($E_p > \hat{E}$) and left-tailed ($E_p < \hat{E}$) prevalence as a function of effect size. Orange lines show the effect size corresponding to the two-sided $\alpha = 0.05$ within-participant test, as used in *Figure 1*. Dashed lines show the effect size corresponding to the ground truth of the simulation. (**A,B**) $\mu_{pop} = 0$, (**C,D**) $\mu_{pop} = 1$. (**A,C**) T = 20 trials, (**B,D**) T = 500 trials. Black line shows maximum a posteriori (MAP), shaded region shows 96% highest posterior density interval (HPDI).

estimation can therefore be interpreted as estimating the population prevalence of participants for which $E_p > \hat{E}$. In the NHST case, $\hat{E}$ is chosen so that $P(E > \hat{E}) = \alpha$ (usually $\alpha = 0.05$), but we can generally consider the prevalence of participants with effects exceeding any value $\hat{E}$. We therefore estimate the prevalence of $E_p > \hat{E}$ as a function of $\hat{E}$. This can reveal if a data set provides evidence for population prevalence of a subthreshold within-participant effect, as well as showing how population prevalence decreases for larger effects. *Figure 5* demonstrates this approach for the simulated systems of *Figure 1*,

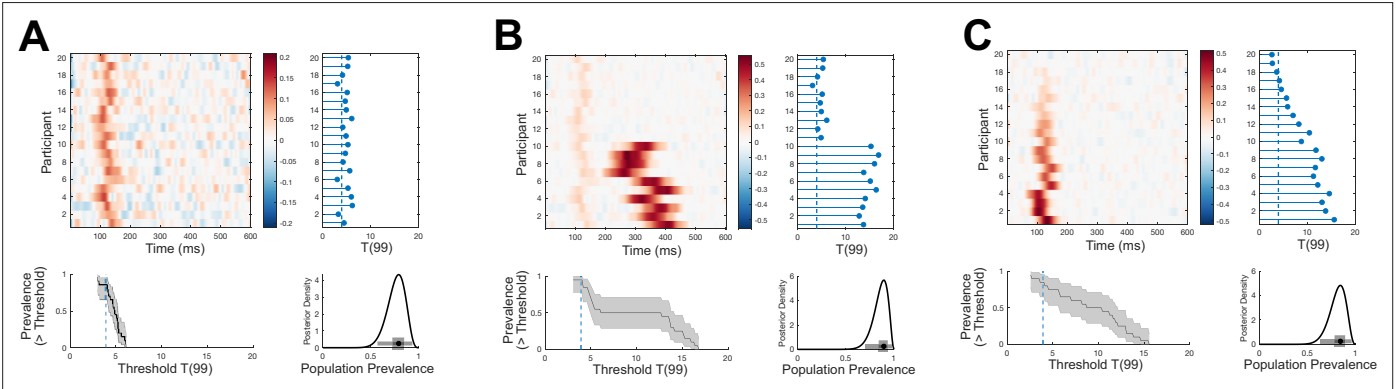

**Figure 6.** Examples of different effect size prevalence curves with similar p=0.05 prevalence. EEG traces are simulated for 100 trials from 20 participants as white noise [$N(0, 1)$] with an additive Gaussian activation ($\sigma = 20$ ms) with amplitudes drawn from a uniform distribution. For each simulation, mean traces are shown per participant (upper-left panel). A within-participant t-test is performed at each time point and for each participant separately (right-hand panel); the blue points show the maximum T-statistic over time for each participant, and the dashed line shows the *p*=0.05 Bonferroni corrected threshold. Lower-right panel shows posterior distribution of population prevalence for an effect in the analysis window. Black curves (lower-left panel) show the prevalence (maximum a posteriori [MAP], shaded area 96% highest posterior density interval [HPDI]) as a function of effect size threshold. (**A**) A weak early effect is simulated in all participants (peak time uniformly distributed 100–150 ms). (**B**) In addition to the same early effect, a stronger, longer ($\sigma = 40$ ms), and more temporally variable later effect is simulated in 10 participants (peak times 250–450 ms). (**C**) Early events are simulated with the same timing as (**A**), but each participant has a different maximum amplitude (participants ordered by effect size). All three simulations have similar prevalence of *p*=0.05 effects, but show differing patterns of prevalence over different effect size thresholds.

showing that prevalence results for both right-sided, $E_p > \hat{E}$, and left-sided, $E_p < \hat{E}$, effect size thresholds. Note that this approach requires the null distribution to be the same for each participant and requires the false positive rate $\alpha$ to be calculated for each effect size considered. It reveals everything we can learn about the population prevalence of different effect sizes from our data set, exploring beyond the binarisation of the within-participant inference. The asymmetry visible in *Figure 5C,D* reflects the positive population mean for those simulations.

In *Figure 6*, three simulated EEG data sets (c.f. *Figure 2*) show how prevalence as a function of effect size can disambiguate situations where the prevalence of *p*=0.05 NHST rejections does not differ. Each panel (A–C) shows a simulation of a different EEG experiment. All three have a similar population prevalence of *p*=0.05 null hypothesis rejections, as shown in the lower-right posterior distributions. However, the prevalence curves as a function of effect size differ in each simulation. For example, in panel A there is no evidence that the population contains individuals with an effect size greater than *T*(99) = 12, whereas the MAP for prevalence of effects greater than 12 is around 50% in panel B. Similarly, the prevalence curve in panel C reflects the larger population variance of the effect in that simulation compared to panel A. While these differences would also be clear from a standard plot of the participant data (e.g. violin or raincloud plot of per-participant maximal effect sizes), the posterior prevalence curves go beyond descriptive visualisations of the specific data sample by quantifying a formal inference to the population, including its associated uncertainty.

## How to apply Bayesian prevalence in practice

As in the simulation of *Figure 1*, a typical population mean inference is often framed as a two-level summary statistic procedure. At the first level, the effect is quantified within each participant (e.g. a difference in mean response between two conditions). At the second level, the population mean is inferred under the assumption that the effect is normally distributed in the population (i.e. based on the mean and standard deviation of the measured effect across participants). Bayesian prevalence is similarly framed as a two-level procedure. At the first level, a statistical test is applied within each participant, the result of which can be binarized via a within-participant NHST (e.g. using a parametric test, as in our simulation, or alternatively using non-parametric permutation methods, independently for each participant), or via an arbitrary effect size threshold $\hat{E}$. At the second level, the binary results from the first level (i.e. the counts of significant participants) are the input to the Bayesian population prevalence computation. To accompany this paper, we provide code in MATLAB, Python, and R to visualise the full posterior distribution of the population prevalence, as well as extract properties, such as the MAP point estimate and HPDIs. We also provide functions to provide Bayesian estimates of the difference in prevalence between two mutually exclusive participant groups to the same test (between-group prevalence difference), as well as the difference in prevalence between two different tests applied to a single sample of participants (within-group prevalence difference). We suggest reporting population prevalence inference results as the MAP estimate together with one or more HPDIs (e.g. with probability 0.5 or 0.96, see Materials and methods).

It is important to stress that the second-level prevalence inference does not impose any requirements on the first-level within-participant tests, other than that each test should provide strong control of the same false positive rate $\alpha$ (see Materials and methods). It is not required, for example, that each participant have the same number of trials, degrees of freedom, or within-participant variance. The within-participant test can be parametric (e.g. a t-test) or non-parametric (e.g. based on permutation testing). It can be a single statistical test or an inference over multiple tests (e.g. a neuroimaging effect within a certain brain region), provided that the family-wise error rate is controlled at $\alpha$ (e.g. by using permutation tests with the method of maximum statistics).

## Effect of number of participants and number of trials

*Figure 7* illustrates how Bayesian prevalence inference scales with the number of participants and trials. *Figure 7A–C* suggests that there are benefits to having larger numbers of participants for the Bayesian prevalence metrics (including decrease in variance of obtained MAP and HPDI width, increase in prevalence lower bound). However, above around 50 participants, these benefits become less pronounced. *Figure 7E* shows that, on average, inferred prevalence is mostly sensitive to the number of trials per participant (horizontal contours) and is invariant to the number of participants (although variance decreases with increasing N, as in *Figure 7A, C and F*). By comparison, t-test

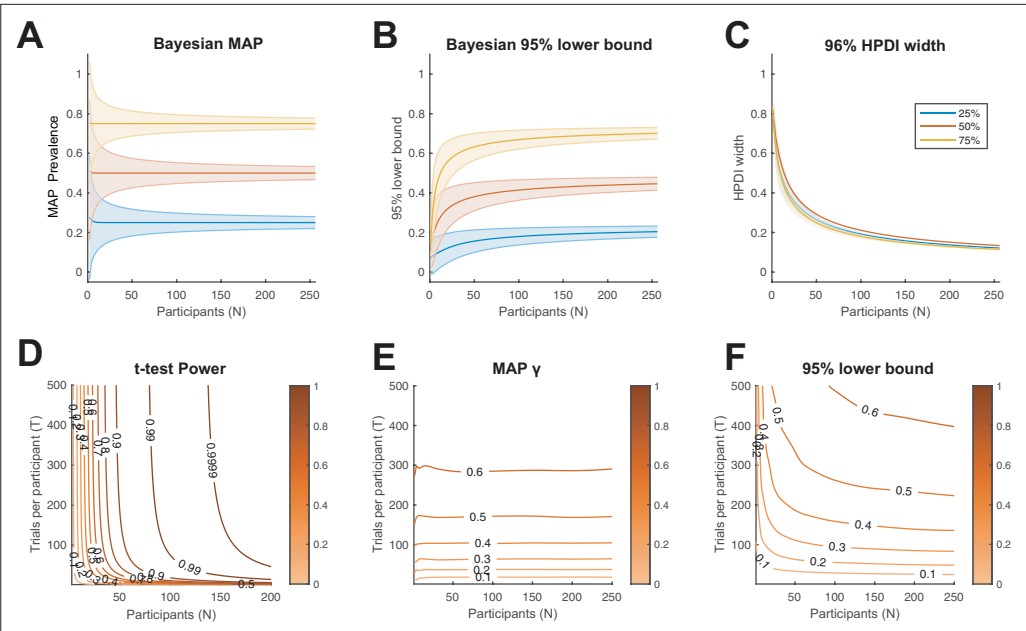

**Figure 7.** Characterisation of Bayesian prevalence inference. (**A–C**) We consider the binomial model of within-participant testing for three ground truth population proportions: 25%, 50%, and 75% (blue, orange, yellow, respectively). We show how (**A**) the Bayesian maximum a posteriori (MAP) estimate, (**B**) 95% Bayesian lower bound, and (**C**) 96% highest posterior density interval (HPDI) width scale with the number of participants. Lines show theoretical expectation, coloured regions show ±1 s.d. (**D–F**) We consider the population model from *Figure 1C and D* ($\mu = 1$). (**D**) Power contours for the population inference using a t-test (*Baker et al., 2020*). Colour scale shows statistical power (probability of rejecting the null hypothesis). (**E**) Contours of average Bayesian MAP estimate for $\gamma$. Colour scale shows MAP prevalence proportion. (**F**) Contours of average 95% Bayesian lower bound for $\gamma$. Colour scale shows lower bound prevalence. From the prevalence perspective, the number of trials obtained per participant has a larger effect on the resulting population inference than does the number of participants.

power (*Figure 7D*) is mostly sensitive to the number of participants (vertical contours) and is largely invariant to the number of trials above around 100 trials per participant (*Baker et al., 2020*). In sum, compared to the population mean t-test, prevalence exhibits greater sensitivity to the number of trials obtained per participant and less sensitivity to the number of participants.

## Discussion

The fields of psychology and neuroimaging are currently dominated by a statistical approach in which the primary focus is the population mean. Here, we propose a conceptual shift in perspective, from estimating population means to estimating the prevalence of effects detected within individuals. We provide a simple but novel Bayesian method to estimate the population prevalence of true positive results from any experimental test. We propose that within-participant inference, combined with prevalence estimation, may often match a researcher's qualitative hypothesis regarding the existence of an effect in the population better than a binary inference that the population mean differs from zero. Particularly in the cognitive sciences, we would argue that effects within individuals are often the ultimate object of study, although these must be assessed at the population level to generalize from a specific experiment. Bayesian prevalence estimation provides this generalisation. It can easily be applied to almost any statistical evaluation of any experiment, provided a NHST can be performed at the individual participant level. The simulations presented here can also be used for simple power analyses when designing studies from this perspective.

Together, this conceptual shift and novel practical method support an alternative perspective for statistics in which the individual participant becomes the most relevant experimental unit to consider for the purposes of replication (*Nachev et al., 2018*; *Smith and Little, 2018*; *Thiebaut de Schotten*

*and Shallice, 2017*), and where power should be considered for effects within individual participants. This perspective gives a very different view of the strength of evidence provided by a data set and of the importance of sample size (for both participants and trials) compared to the more common population mean perspective (*Baker et al., 2020*). For example, the simulation of 50 participants with 20 trials in *Figure 1C* has $p=0.008$ for a group mean different from zero, a result that is as surprising, under the null hypothesis, as observing seven heads in a row from tosses of a fair coin (*Rafi and Greenland, 2020*). This is weaker evidence than just two out of five participants showing an effect at $\alpha = 0.05$ ($p=0.0012$ under the global null, or about as surprising as 10 heads in a row). Three out of five significant participants correspond to $p=0.00003$ under the global null (as surprising as 15 heads in a row); this is substantially stronger evidence for a population-level effect than that provided by the population mean inference in *Figure 1D* (from 50 participants, even with 500 trials). However, in the current scientific climate, the weaker result obtained from the larger sample size would commonly be viewed as providing more satisfactory evidence by most readers and reviewers. We would like to highlight this pervasive misunderstanding: that larger participant numbers automatically imply better evidence at the population level. The crux of our argument is that most studies focus on the difference between *Figure 1A and C* (a small difference in population mean, ignoring the implications of large between-participant variance in studies that are underpowered for within-participant effects). In contrast, moving towards the situation shown in *Figure 1B and D* (increased power within individual participants) would provide both improved replicability and greater insight into the size and prevalence of effects in the population.

The practice of collecting enough data to perform within-participant inference is not a new idea—much of traditional psychophysics employs this approach (*Smith and Little, 2018*). We have employed this technique with EEG (*Schyns et al., 2011*; *Smith et al., 2006*) and MEG (*Ince et al., 2015*; *Zhan et al., 2019*), and in fMRI it is becoming more common to collect large quantities of data for fitting high-dimensional, cross-validated machine learning models within individual participants (*Huth et al., 2016*; *Naselaris et al., 2021*; *Stansbury et al., 2013*). Recently, this practice has also been adopted in the resting-state fMRI field where it is termed *dense sampling* or *precision imaging* (*Braga and Buckner, 2017*; *Gordon et al., 2017*; *Laumann et al., 2015*; *Poldrack, 2017*), or more recently *deep imaging* (*Gratton and Braga, 2021*). Ensuring that experiments are sufficiently powered to obtain reliable effect size estimates within individual participants could benefit studies relating individual differences in experimental effects to other external factors. In fact, the focus on population mean may have emphasised experimental effects which have low between-participant variance and which are therefore less well-suited for studying individual differences (*Elliott et al., 2020*; *Hedge et al., 2018*). With more reliable within-participant estimates, comparisons between groups can look beyond means to consider other differences in the distribution of effect sizes (*Rousselet et al., 2017*), as we show here considering prevalence as a function of effect size (*Figure 6*).

Note that while similar points regarding within-participant inference have been made elsewhere (*Grice et al., 2020*; *Nachev et al., 2018*; *Smith and Little, 2018*; *Thiebaut de Schotten and Shallice, 2017*), such results are typically considered to form a case study, without an inferential link that allows generalisation to the population (*Neuroscience, S. for, 2020*). ('[The within-participant] approach only allows for statements that pertain to the existence and magnitude of effects in those subjects, rather than in the populations those subjects are drawn from.') The methods presented here address this concern by providing an inferential bridge from within-participant tests to the population level, even when the number of participants is small. Randomisation tests (*Edgington et al., 2007*) provide a completely non-parametric and assumption-free approach to experimental statistics, but their interpretation is limited since they cannot provide statistical statements which generalise to a population. The combination of within-participant randomisation tests with Bayesian prevalence provides a population inference with minimal assumptions (compared to tests of population mean which usually assume a normal distribution).

Bayesian prevalence produces a quantitative prevalence estimate with associated uncertainty, rather than reducing an entire experiment to a single, binary, population NHST result. This addresses many of the problems noted with the NHST framework (*Amrhein et al., 2019*; *McShane et al., 2019*) and also reduces the risk of questionable research practices, such as p-hacking (*Forstmeier et al., 2017*). Each participant serves as a replication since the same statistical analysis is performed on separate sets of data. This provides a degree of protection from problems such as researcher degrees

of freedom. While it takes on average 20 different independent analyses of null data to find a single $p=0.05$ population mean NHST rejection in a multi-participant data set, it would take 400 different analyses to reject a within-participant NHST at $p=0.05$ by chance in two different participants, or 8000 different analyses to find three participants significant by chance. Therefore, within-participant inference provides exponential protection with increasing participants against issues such as p-hacking and researcher degrees of freedom, likely resulting in more robust and replicable findings. We have also shown how to estimate the prevalence of different effect size thresholds, which avoids focusing on a single within-participant dichotomisation. However, estimated prevalence is only one approach to evaluate the strength of evidence provided by a data set, and it should be assessed alongside the quality of the experimental design, within-participant power, and the effect sizes within individuals. Prevalence and population mean estimation are not mutually exclusive; ideally one could report within-participant effect sizes and estimated population prevalence together with estimation and inference of the population mean.

It is also possible to perform frequentist inference on the prevalence testing model (*Allefeld et al., 2016*; *Donhauser et al., 2018*). Various null hypotheses for specific prevalence values can be considered, for example, the global null ($\gamma = 0$), the majority null ($\gamma < 0.5$), or any value of interest. However, this approach results in a binarisation of the experimental outcome to significant or not, and adds a degree of freedom to the researcher in terms of the particular value of $\gamma$ to test. An alternative approach is to consider the family of null hypotheses $\gamma < \gamma_c$ and find the largest $\gamma_c$ value for which the corresponding null hypothesis can be rejected at the chosen $\alpha$. This can be interpreted as a frequentist lower bound on the true prevalence given the specified false positive rate. However, the Bayesian posterior we present here gives a full quantitative description of the population based on the evidence provided by the data, avoids dichotomisation of the final analysis result, provides direct comparison between different groups or tests, and allows for prevalence as a function of effect size to give a more detailed picture of the population. Prevalence as a function of effect size can give insights into the variation of effects across the population (e.g. subgroups, *Figure 6B*) without requiring a parametric model (e.g. a normal distribution).

We have focused on human participants in typical psychology or neuroimaging experiments, but the proposed methods can be applied to infer the population prevalence of effects in other types of experimental units. For example, our approach could be directly applied to identified single-unit neurons in electrophysiological recordings. Typically, only a subset of the recorded neurons respond to a specific stimulus or experimental manipulation. Bayesian prevalence can be applied to estimate the proportion of neurons that respond in a particular way in the population of neurons in a particular brain region. For example, one could consider well-isolated single units recorded in the ventroposterior medial nucleus (VPm) of rodent thalamus (*Petersen et al., 2008*). Of all the *N* cells recorded (across sessions and perhaps animals), a certain number *k* might be found to be sensitive to the velocity of whisker motion, based on a non-parametric test of an encoding model. Applying Bayesian prevalence to this data would give a posterior distribution for the proportion of neurons in VPm thalamus that are velocity sensitive in a particular experimental preparation. Our between-group comparison method could also be used to formally compare the population prevalence of a certain type of responsive neuron between different brain areas, between different species, or between different experimental conditions (e.g. before or after learning, or with or without a pharmacological intervention). Thus, although it is common in electrophysiology to have individual neurons as the replication unit and perform inference at that level, the inferential bridge to a population of cells that Bayesian prevalence provides offers a new perspective on the results of such studies.

One potential criticism is that the demonstration of within-participant effects, which the prevalence approach requires, sets a much higher bar of evidence. It might be impractical to reach sufficient within-participant power in some imaging modalities with typical experimental designs. However, many statisticians have explicitly argued that the replicability crisis results from standards of evidence for publication being too weak (*Benjamin et al., 2018*). If so, the within-participant approach should lead to increased replicability. Indeed, it is common in neuroimaging studies to have no consistent pattern discernible in the effect size maps of individual participants, and yet such studies report a significant population mean effect in a focal region. In our view, this is problematic if our ultimate goal is to relate neuroimaging results to cognitive functions within individuals (*Naselaris et al., 2021*; *Smith and Little, 2018*). By contrast, as our simulations show (*Figures 1B and 2*), strong evidence for

a modulation can occur in the absence of a population mean effect when the effect is heterogeneous across participants.

It is natural to expect considerable heterogeneity to exist across populations for a wide range of experimental tasks. In fact, the normal distribution that underlies most inferences on the population mean implies such heterogeneity (*Figure 1B*). In clinical studies of rare diseases, patient numbers are often limited, and heterogeneity can be higher relative to the healthy population. If an experiment is sufficiently powered within individual participants, then Bayesian prevalence provides a statistical statement about the population of patients with the disease, even from a small sample and without assuming a normal distribution for the population.

It has recently been suggested that researchers should define a smallest effect size of interest (SESOI) (*Lakens, 2017*) and should consider this when calculating statistical power. We suggest that the SESOI should also be considered at the individual level and explicitly related to the population variance obtained from hierarchical mixed effects models. If the population variance is large, then this implies that individuals exist in the population who have effects that are larger than the individual SESOI, even if the population mean is not larger. This possibility is almost always ignored in applications of such modelling, which usually focus only on the population mean. Equivalence tests (*Lakens, 2017*) could be applied within each participant and used to estimate the population prevalence of effects significantly smaller than the SESOI (i.e. the population prevalence of the absence of the effect).

Furthermore, the common assumption that effect sizes in the population follow a normal distribution is strong, although seldom justified (*Lakens et al., 2018*). For example, information processing, decision making, risk taking, and other strategies might vary categorically within the healthy population and across clinical and sub-clinical groups (*Nachev et al., 2018*; *Smith and Little, 2018*). In neuroimaging studies, there are related issues of variability in anatomical and functional alignment. To address these issues, results are often spatially smoothed within individuals before performing population mean inference. However, many new experimental developments, such as high-resolution 7T fMRI to image cortical layers and high-density intracranial recordings in humans, pose increasing difficulties in terms of aligning data across participants. A major advantage of the prevalence approach is that we can perform within-participant inference corrected for multiple comparisons, and then estimate the population prevalence of these effects without requiring them to be precisely aligned (in space, time frequency, etc.) across participants (as illustrated in *Figure 2*). For example, one might report that 24/30 participants showed an EEG alpha band power effect between 500 ms and 1000 ms post stimulus, which implies a population prevalence MAP of 0.79 (96% HPDI [0.61 0.91]), without requiring these individual effects to occur at precisely the same time points in those 24 participants. Similarly, if within-participant inference is corrected for multiple comparisons, one can infer prevalence of, say, an effect in a certain layer of primary visual cortex, without requiring precise alignment of the anatomical location of the effect between participants (*Fedorenko, 2021*).

Of course, there are many cases where the population mean is indeed the primary interest; in such cases, estimating and inferring on the population mean using hierarchical models is the most appropriate analysis, when possible. Linear mixed-effect models can also be interrogated in different ways to investigate the question of prevalence, for example, by examining the variance of the by-participant random slopes or explicitly computing prevalence of different effect sizes from the normal distribution fit to the population. It is possible to extend Bayesian hierarchical models to explicitly account for different sub-groups of participants (*Bartlema et al., 2014*; *Haaf and Rouder, 2017*; *Haaf and Rouder, 2019*; *Rouder and Haaf, 2020*). However, these approaches currently are not widely adopted, cannot easily be applied to non-linear or high-dimensional analysis methods common in neuroimaging, and add both mathematical and computational complexity compared to the second-level Bayesian prevalence method we present here, which is straightforward to apply to any first-level, within-participant analysis. Further, for many complex computational techniques—from modelling learning behaviour and decision making, to neuroimaging analysis techniques such as Representational Similarity Analysis, or high-dimensional encoding models (*Haxby et al., 2014*)—it is currently not possible to employ a multi-level linear modelling approach down to the trial level due to the complexity of the non-linear analysis functions and models employed. It is also important to note that if the effect of interest is an unsigned information measure such as mutual information (*Ince et al., 2017*) or classification accuracy (*Haxby et al., 2014*) then the t-test on the population mean does not

actually provide a random-effects inference that generalises to the population, but is equivalent to rejection of the global null showing only some sampled participants have an effect, that is, a fixed-effect analysis (*Allefeld et al., 2016*). In such cases, Bayesian prevalence estimates may be more descriptive and provide stronger generalisation than the population mean NHST.

In sum, Bayesian prevalence has a broad range of applicability—spanning dense sampling studies with high within-participant power, to more traditional sampling models (more participants, fewer trials, e.g. *Figure 1C*). It is applicable to any behavioural study, including detailed computational models of behaviour, provided that model comparison or inference on model parameters can be performed within individuals. In neuroimaging, Bayesian prevalence can be applied to any imaging modality (EEG, MEG, fMRI, fNIRS, intracranial EEG), to individual neurons within a brain region (to infer the proportion of responsive neurons), and with any statistical approach, including non-linear and multivariate analysis methods. The crucial requirement is an adequately powered experiment to detect effects within individual participants (or other units of interest, e.g. neurons). We argue that ensuring experiments are adequately powered to detect effects within individuals would have a wide range of advantages, including strengthening the robustness and replicability of reported results.

## Conclusions

While the problems that underlie the replication crisis are being increasingly recognised, there is currently no consensus as to alternative statistical approaches that are needed to address the underlying problems. Here, we propose that shifting our focus to quantifying and inferring effects within individuals addresses many of the pressing concerns recently highlighted in psychology and neuroimaging (*Amrhein et al., 2019*; *Benjamin et al., 2018*; *Forstmeier et al., 2017*; *Ioannidis, 2005*; *McShane et al., 2019*). We present a Bayesian approach to estimating population prevalence which is broadly applicable as it places no assumptions on the within-participant tests nor on the distribution of effects in the population. Further, prevalence does not require a Bayesian treatment; frequentist inference approaches can be used instead. The crucial point is to shift our perspective to first evaluate effects within individual participants, who represent the natural replication unit for studies of brains and behaviour.

## Materials and methods
### Simulations from an hierarchical population model

The data shown in *Figure 1* were simulated from the standard hierarchical model:

$$
\begin{aligned}
y_{ij} &\sim N\left(\mu_i,\ \sigma_w^2\right) \\
\mu_i &\sim N\left(\mu_{pop},\ \sigma_b^2\right)
\end{aligned}
$$

where $y_{ij}$ denotes the measurement made on the $j$th trial (out of $T$) of the $i$th participant (out of $N$). $\mu_i$ represents the mean of each individual participant, $\sigma_w$ represents a common within-participant standard deviation over trials, $\sigma_b$ represents the standard deviation between participants, and $\mu_{pop}$ represents the overall population mean. This can be written as

$$
y_{ij} = \mu_{pop} + \eta_{ij} + \epsilon_i
$$

where $\eta_{ij} \sim N\left(0, \sigma_w^2\right)$, and $\epsilon_i \sim N\left(0, \sigma_b^2\right)$. Note that under this model the distribution of the within-participant means is $N\left(\mu_{pop}, \sigma_b^2 + \frac{1}{t}\sigma_w^2\right)$.

### Binomial model of population prevalence of the ground truth state of sampled units

We consider a population of experimental units (e.g. human participants or individual neurons) which are of two types: those that have a particular binary effect or property, and those that do not. We consider the population prevalence of the ground truth state of each unit $\gamma_{gt}$, which is the proportion of the population from which the sample was drawn that have the effect ($0 < \gamma_{gt} < 1$). If the true status of each individual unit could be directly observed, then the sample could be modelled with a binomial distribution with probability parameter $\gamma_{gt}$. However, we cannot directly observe the true

status of each unit. Instead, we apply to each unit a statistical test following the NHST framework, which outputs a binary result (which we term here positive vs. negative). This test has a false positive rate $\alpha$, and sensitivity $\beta$. Thus, the probability that a randomly selected unit from the population that does not possess the defined effect but will produce a positive test result is $\alpha$, while the probability that a randomly selected unit that does possess the defined effect will produce a positive test result is $\beta$. Under the assumption that the units are independent and $\alpha$ and $\beta$ are constant across units, the number of positive tests $k$ in a sample of size $n$ can be modelled as a binomial distribution with parameter $\theta$ (*Donhauser et al., 2018*; *Friston et al., 1999b*; *Friston et al., 1999a*; *Rogan and Gladen, 1978*):

$$P(X = k|\theta) = \binom{n}{k} \theta^k (1 - \theta)^{n-k}$$
$$\theta = \left(1 - \gamma_{gt}\right) \alpha + \gamma_{gt} \beta$$

## Binomial model of population prevalence of true positives for a given test procedure

A major issue with the above approach is that it requires the sensitivity $\beta$ to be specified and constant across units. $\beta$ is the probability of a significant result given that the individual has an effect and differs as a function of the ground truth effect size. In general, we do not assume that all members of the population have the same effect size, so it is not possible to specify a common $\beta$ for all individuals.

Therefore, rather than modelling the prevalence of ground truth effects, we consider the prevalence of true positive test results for the particular test procedure we employ, $\gamma_{tp}$:

$$P(X = k|\theta) = \binom{n}{k} \theta^k (1 - \theta)^{n-k}$$
$$\theta = \left(1 - \gamma_{tp}\right) \alpha + \gamma_{tp}$$

In this case, the only assumption is that test procedure has the same false positive rate $\alpha$ for every individual, which is easily satisfied by most common parametric and non-parametric statistical tests. Note that this is equivalent to estimating ground truth prevalence with a test with $\beta = 1$. In general, a test with lower sensitivity allows inference of a higher prevalence for an observed $k$ because some of the observed negative results will be missed true positive results. Therefore, prevalence of true positives obtained with $\beta = 1$ is a conservative lower bound on the prevalence of ground truth state (*Allefeld et al., 2016*; *Donhauser et al., 2018*; *Friston et al., 1999a*).

Throughout this paper we consider the prevalence of true positives for a given test procedure rather than the prevalence of ground truth state and so omit the subscript *tp*. This quantifies the proportion of the population that would be expected to provide a true positive test result (i.e. have a non-null effect that would be detected by the experimental test procedure considered).

## Frequentist estimation of and inference on population prevalence

Various frequentist approaches can be used with the above binomial model of statistical testing. First, the maximum likelihood estimate of the population prevalence of true positives can be obtained as

$$\hat{\gamma} = \frac{k/n - \alpha}{1 - \alpha}$$

Standard bootstrap techniques (*Johnson, 2001*) can give percentile bootstrap confidence intervals as an indication of uncertainty in this estimate. We can also explicitly test various null hypotheses at the population level. For example, we can test a compound null hypothesis $\gamma < 0.5$, termed the majority null (*Allefeld et al., 2016*; *Donhauser et al., 2018*). This is chosen with the idea that a prevalence of >50% supports a claim that the effect is *typical* in the population. Other explicit compound nulls of this form can also be tested (e.g. that $\gamma < 0.25$ or $\gamma < 0.75$). Alternatively, it is possible to infer a lower bound on the population prevalence by finding the largest value $\gamma_c$, such that $p\left(X > k \,|\, \gamma < \gamma_c\right) < 0.05$ (*Allefeld et al., 2016*; *Donhauser et al., 2018*). This inferred lower bound provides a more graded

output than a binary significance result of testing against a specific compound null (i.e. the continuous value $\gamma_c$).

## Bayesian estimation of population prevalence

We apply standard Bayesian techniques to estimate the population prevalence parameter of this model (**Gelman, 2014**). Assuming a beta prior distribution for $\theta$ with shape parameters $r, s$, together with a binomial likelihood function, the posterior distribution for $\theta$ is given by a beta distribution with parameters $(k + r, \; n - k + s)$, truncated to the interval $[\alpha, 1]$, where $k$ is the number of participants showing an above-threshold effect out of $n$ tested. In the examples shown here, we use a uniform prior (beta with shape parameters $r = s = 1$), as in the general case there is no prior information regarding $\theta$. This implies a uniform prior also for $\gamma$, so, a priori, we consider any value of population prevalence equally likely. While we default to the uniform prior, the code supports any beta distribution as a prior. Alternative priors could be implemented via Markov chain Monte Carlo methods (**Gelman, 2014**) together with the models described here. Note that similar Bayesian approaches have been applied in the field of epidemiology, where sometimes multiple complementary diagnostic tests for a disease are applied with or without a gold standard diagnosis in a subset of the sampled units (**Berkvens et al., 2006**; **Enøe et al., 2000**; **Joseph et al., 1995**).

Under the uniform prior, the Bayesian MAP estimate for prevalence proportion of true positives is available analytically and is equivalent to the maximum likelihood estimate:

$$\gamma_{map} = \frac{k/n - \alpha}{1 - \alpha}$$

Following **McElreath, 2016**, we present 96% HPDIs here to emphasise the arbitrary nature of this value and reduce the temptation to interpret the interval as a frequentist $p$=0.05 inference.

## Bayesian estimation of the prevalence difference between two independent groups

We consider here a situation where the same test is applied to units sampled from two different populations. In addition to the prevalence of true positives within each population, we wish to directly estimate the difference in prevalence between the two populations. We denote the prevalence within each population as $\gamma_1, \gamma_2$, respectively. We sample $n_1, n_2$ participants at random from each population and record $k_1, k_2$, the number of significant within-participant tests in each sample. Assuming independent uniform priors on the prevalences and associated $\theta_i$ variables as above with:

$$\theta_i = (1 - \gamma_i)\,\alpha + \gamma_i$$

then the posterior distribution for $(\theta_1, \theta_2)$ is given by the product of two truncated beta distributions, with parameters $(k_i + 1, \; n_i - k_i + 1)$, respectively, both truncated to the interval $[\alpha, 1]$. The prevalence difference can be obtained as:

$$\gamma_1 - \gamma_2 = (\theta_1 - \theta_2) / (1 - \alpha)$$

For non-truncated beta distributions, an analytic exact result is available (**Pham-Gia et al., 1993**). This result could be extended to provide an exact distribution for the prevalence difference, but the mathematical expressions involved are quite complex. We find it simpler to employ Monte Carlo methods, which can provide as close an approximation to the exact answer as desired. Here we use Monte Carlo methods to draw samples from the posterior for $(\theta_1, \theta_2)$, obtaining samples for the prevalence difference with the above expression. We use these samples to numerically compute the MAP and HPDIs.

## Bayesian estimation of the prevalence difference of two different tests within the same sample of participants

In this situation, we consider that two different test procedures are applied to a single sample of $n$ units. We assume both tests have the same value of $\alpha$ and define:

$$\theta_i = (1 - \gamma_i)\,\alpha + \gamma_i = \alpha + (1 - \alpha)\,\gamma_i$$

Here $\theta_i$ is the probability that a randomly selected unit from the population will show a positive result on the $i$th test, and $\gamma_i$ is the population prevalence of true positives associated with the $i$th test. Now, each unit provides one of four mutually exclusive results based on the combination of binary results from the two tests. $k_{11}$ represents the number of units that have a positive result in both tests, $k_{10}, k_{01}$ represent the number of units that have a positive result only in the first or second test, respectively, and $k_{00}$ is the number of units that do not show a positive result in either test. So $\sum_{i,j} k_{ij} = n$. We can define analogous variables $\boldsymbol{\theta} = \{\theta_{ij}\}$, representing the population proportion for each of the four combined test outcomes. Note that $\theta_{ij} > 0$ and $\sum_{i,j} \theta_{ij} = 1$. The marginal success probabilities $\theta_i$ can be expressed as:

$$\theta_1 = \theta_{11} + \theta_{10}, \quad \theta_2 = \theta_{11} + \theta_{01}$$

and so

$$\gamma_1 - \gamma_2 = \left(\theta_{10} - \theta_{01}\right) / \left(1 - \alpha\right)$$

The marginal probabilities $\theta_i$ are subject to the constraints

$$\alpha < \theta_i < 1$$

and so

$$\alpha < \theta_{11} + \theta_{10} < 1, \quad \alpha < \theta_{11} + \theta_{01} < 1$$

Assuming a uniform prior and a multinomial distribution for the $k_{ij}$, the posterior of $\boldsymbol{\theta}$ is a truncated Dirichlet distribution with parameters $m_{ij} = k_{ij} + 1$ subject to the constraints above (which are analogous to the truncation of the beta posterior distribution in the case of a single test). We use a Monte Carlo approach to draw samples from this posterior following a modified stick-breaking process.

- Draw a sample $\theta_{11}$ from a $Beta\left(m_{11}, m_{10} + m_{01} + m_{00}\right)$ distribution truncated to the interval $[0,1]$.
- Draw a sample $z_{10}$ from a $Beta\left(m_{10}, m_{01} + m_{00}\right)$ distribution truncated to the interval $\left[\max\left(\frac{\alpha - \theta_{11}}{1 - \theta_{11}}, 0\right), \frac{1 - \theta_{11}}{1 - \theta_{11}}\right]$. Set $\theta_{10} = \left(1 - \theta_{11}\right) z_{10}$.
- Draw a sample $z_{01}$ from a $Beta\left(m_{01}, m_{00}\right)$ distribution truncated to the interval $\left[\max\left(\frac{\alpha - \theta_{11}}{1 - \theta_{11} - \theta_{10}}, 0\right), \min\left(\frac{1 - \theta_{11}}{1 - \theta_{11} - \theta_{10}}, 1\right)\right]$. Set $\theta_{01} = \left(1 - \theta_{11} - \theta_{10}\right) z_{01}$.
- Set $\theta_{00} = 1 - \theta_{11} - \theta_{10} - \theta_{01}$.
- Then $\left(\theta_{11}, \theta_{01}, \theta_{10}, \theta_{00}\right)$ is a draw from the required truncated Dirichlet distribution, and $\left(\theta_{10} - \theta_{01}\right) / \left(1 - \alpha\right)$ is a draw from the posterior distribution of the prevalence difference.

We use these samples to numerically compute properties like the MAP estimate and HPDIs.

To specify a ground truth to simulate data from two tests applied to the same participants (**Figure 3**), we require $\gamma_1$ and $\gamma_2$, the population prevalences of the two tested effects, together with $\rho_{12}$, the correlation between the presence of the two effects across the population. From this we can calculate $\gamma_{11}$, the proportion of true positive results to both tests as

$$\gamma_{11} = \gamma_1 \gamma_2 + \rho_{12} \sqrt{\gamma_1 \left(1 - \gamma_1\right) \gamma_2 \left(1 - \gamma_2\right)}$$

Similarly, we can define $\gamma_{ij}$ representing the population proportions corresponding to the other test result configurations. Then we can generate multinomial samples using the parameters $\theta_{ij}$ computed as

$$\theta_{11} = \gamma_{11} + \alpha^2 \gamma_{00} + \alpha \gamma_{01} + \alpha \gamma_{10}$$
$$\theta_{10} = \alpha + (1 - \alpha)\gamma_1 - \theta_{11}$$
$$\theta_{01} = \alpha + (1 - \alpha)\gamma_2 - \theta_{11}$$
$$\theta_{00} = 1 - \theta_{11} - \theta_{10} - \theta_{01}$$

## Prevalence as a function of effect size threshold

Estimating the prevalence of $E_p > \hat{E}$ proceeds as for prevalence inference based on within-participant NHST. One additional step is the need to calculate $\alpha$, the false positive rate under the null hypothesis

of no effect, for each threshold value $\hat{E}$. This is simply the probability of $E_p > \hat{E}$ under the null hypothesis. In the examples shown here, we calculate this from the cumulative distribution function of the appropriate t-distribution, but for other tests this could also be estimated non-parametrically. A number of $\hat{E}$ values are selected, linearly spaced over the observed range of the sample. For each of these values the count of the number of participants satisfying the inequality and the $\alpha$ value corresponding to the inequality are used to obtain the Bayesian posterior for prevalence of true positives. Note that this can be applied to either tail $E_p > \hat{E}$ or $E_p < \hat{E}$.

### Data availability

Scripts implementing the simulations and creating the figures from the paper are available at https://github.com/robince/bayesian-prevalence (copy archived at swh:1:rev:a10f2760930f7638d1c2a-73944719e6283aedcec, *Ince, 2021*).

### Code availability

To accompany this paper, we provide functions in MATLAB, Python, and R to calculate the Bayesian prevalence posterior density (e.g. to plot the full posterior distribution), the MAP estimate of the population prevalence, HPDI intervals of the posterior and lower bound quantiles of the posterior, as well as prevalence differences between samples or between tests within a sample. We also provide example scripts that produce posterior plots as in *Figure 1E*. See https://github.com/robince/bayesian-prevalence.

### Acknowledgements

We thank Carsten Allefeld for useful discussions. RAAI was supported by the Wellcome Trust [214120/Z/18/Z]. PGS was supported by the EPSRC [MURI 1720461] and the Wellcome Trust [107802]. PGS is a Royal Society Wolfson Fellow [RSWF\R3\183002].

## Additional information

### Funding

| Funder | Grant reference number | Author |
| --- | --- | --- |
| Wellcome Trust | 214120/Z/18/Z | Robin AA Ince<br>Angus T Paton |
| Wellcome Trust | 107802 | Philippe G Schyns |
| Engineering and Physical Sciences Research Council | MURI 1720461 | Philippe G Schyns |
| Royal Society | RSWF\R3\183002 | Philippe G Schyns |

The funders had no role in study design, data collection and interpretation, or the decision to submit the work for publication.

### Author contributions

Robin AA Ince, Conceptualization, Formal analysis, Funding acquisition, Investigation, Methodology, Project administration, Software, Visualization, Writing – original draft, Writing – review and editing; Angus T Paton, Data curation, Formal analysis, Validation, Visualization, Writing – review and editing; Jim W Kay, Conceptualization, Formal analysis, Investigation, Methodology, Software, Writing – review and editing; Philippe G Schyns, Conceptualization, Funding acquisition, Writing – review and editing

### Author ORCIDs

Robin AA Ince (iD) http://orcid.org/0000-0001-8427-0507

### Decision letter and Author response

Decision letter https://doi.org/10.7554/eLife.62461.sa1
Author response https://doi.org/10.7554/eLife.62461.sa2

## Additional files

### Supplementary files
• Transparent reporting form

### Data availability
Scripts implementing the simulations and creating the figures from the manuscript are available at https://github.com/robince/bayesian-prevalence (copy archived at https://archive.softwareheritage.org/swh:1:rev:a10f2760930f7638d1c2a73944719e6283aedcec). A toolbox of functions to calculate Bayesian prevalence is provided for Matlab, Python and R.

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
