## [Decision Letter]

**Acceptance summary:**

This is paper describes methods to move away from frequentist null hypothesis testing toward a Bayesian approach of estimate the prevalence of within-participant effects. This methodological advance should be widely applicable in many fields, and should promote a more robust evaluation of effects.

**Decision letter after peer review:**

Thank you for submitting your article "Bayesian inference of population prevalence" for consideration by *eLife*. Your article has been reviewed by 2 peer reviewers, and the evaluation has been overseen by a Reviewing Editor and Timothy Behrens as the Senior Editor. The following individuals involved in review of your submission have agreed to reveal their identity: Alex Huth (Reviewer #1); Sam Ling (Reviewer #2).

The reviewers have discussed the reviews with one another and the Reviewing Editor has drafted this decision to help you prepare a revised submission. We will all re-evaluate the manuscript after the re-submission, and I hope the comments below are useful.

Summary:

As you'll see below, the reviewers and I were all enthusiastic about the paper. That said, we all felt that a revision is required to provide concrete examples of when this method is preferred over frequentist approaches, as well as the general types of questions that this method is appropriate for addressing.

In addition, we all felt that the focus of the paper should be expanded to give more attention to non-fMRI studies. For example, there is passing mention in the Discussion section about how this might apply to single unit studies, but that seems more of an afterthought at the moment. Thus, the paper is missing a chance to speak to, and influence, a much broader audience.

Essential revisions:

In addition to revising the text as indicated above (and in the more detailed reviews), it seems likely that some additional simulations may need to be run to address all of the points. This is particularly true of the suggestion to demonstrate a situation where prevalence estimation is more powerful than standard methods.

*Reviewer #1 (Recommendations for the authors):*

In this paper, Ince and colleagues describe a novel set of Bayesian methods for estimating and working with "population prevalence" of an effect, which they pose as an alternative to NHST on the mean effect in the population. In turns this paper was tantalizing, exciting, and frustrating. I very much want the method the authors propose to be useful, but I was left without a clear sense of (1) what specific questions ("qualitative hypotheses") this method is appropriate for answering, and (2) why (quantitatively) this method should be preferred over others. Specific comments and suggestions are enumerated below.

1. Please show us how prevalence estimation can be more useful than classical approaches! The authors convincingly argued that estimating prevalence amounts to replicating an experiment in each individual subject, a shift in perspective that may help alleviate issues of non-replicability in psychology and neuroimaging (I am very sympathetic to these arguments!). However, it was somewhat frustrating that none of the computational experiments seemed to directly contrast prevalence estimation vs. population mean estimation. If prevalence is indeed a superior way to conceptualize effects in a population, then there should be some experiment where prevalence and population mean give different results for testing the same qualitative hypothesis. For example, in the discussion the authors bring up the issue of uninteresting heterogeneity across subjects, which can reduce the effectiveness of standard approaches (i.e. where the precise anatomical location or timing of an effect is variable across subjects). This paper would be much more convincing if it could experimentally demonstrate a situation like this (in simulation, of course), where prevalence estimation might truly be more powerful than standard methods.

2. Tell us more about what inferences can actually be made using prevalence estimation! The experiment in Figure 1 lays out the differences between prevalence and population mean estimation quite clearly, but the conclusions are much less clear. The authors "suggest that often, if the goal of the analysis is to infer the presence of a causal relationship within individuals in the population, the within-participant perspective may be more appropriate" – but more appropriate for what? Exploring-specifically at this point in the paper-which types of qualitative hypotheses (or some specific examples?) can be meaningfully addressed using prevalence analyses would, in my opinion, greatly strengthen the authors' case.

3. The argument that prevalence estimation is more sensitive to trials per participant than number of participants (Figure 2E) seems circular. The authors define prevalence as the fraction of subjects with "true positive (significant) test results" rather than the fraction where the null hypothesis is actually false, because they can only find a lower bound on the latter measure. Given this definition, the finding that "prevalence greater sensitivity to the number of trials per participant, and less sensitivity to the number of participants" (which is seemingly framed as the main conclusion of the analyses in Figure 2?) is highly unsurprising.

4. How does estimating group differences in prevalence compare to estimating differences in group means? The experiment shown in Figure 3 is tantalizing, but what I really wanted to know here is whether (and under what conditions) comparing groups using prevalence might be more powerful than comparing groups using traditional models.

5. The analysis of prevalence as a function of effect size is missing context and conclusions. Ultimately, I was left without any sense of the utility of estimating prevalence for different effect sizes after reading this section (and Figure 4). What is this specific procedure useful for? What kinds of hypotheses can it test? How does it compare to other methods? Answering these questions would undoubtedly strengthen the paper.

*Reviewer #2 (Recommendations for the authors):*

This manuscript advocates for a shift in the statistical (and methodological) approaches taken in neuroscience and psychology, wherein effects are considered from the perspective of prevalence, rather than population-centric NHST. They provide simple yet compelling simulations to make their case, and introduce the bayesian prevalence method, along with code to implement on one's own. This manuscript is incredibly well written, the simulations are tightly constructed, and will, I hope, become widely read by the neuroscience and psychology community. It's a straightforward treatment of a statistic that the field needs to consider shifting towards, and the ability to generalize to any NHST will no doubt make this user friendly for most all researchers. I will readily admit that this was a method that my own lab has been in search of for a while, and we plan to deploy this in our own work. Below, I outline my questions/comments, but they are largely meant to support clarification to a broader audience.

1. While the authors promote a Bayesian treatment of the estimate of prevalence, they readily admit that a frequentist approach is suitable as well. It would be worth detailing a bit more the advantages and disadvantages of the two, if any. Particularly because the frequentist approach to neuroimaging has been out in the wild for a while now.

2. The authors simulate individuals within the population as having equal within-subject (or unit) variability. To what degree are the assumptions for prevalence estimation upheld in the case where individual subjects have unequal variance?

3. I assume this rests on the assumption of independence between units of replication (ie subjects, voxels, ROIs, cells). Is this true? And how robust is this method to a lack of independence? I ask because if one were carrying this method out on a voxel-by-voxel basis, within a participant, there's quite a bit of non-independence between these voxels. And likely same goes for between adjacent ROIs, or even between time points. It would be helpful if the authors dedicated some text to this assumption.

4. This comment is more to aid in promotion of the work, but I think the introduction and walkthrough of the simulation would be better serviced by using a case example throughout. It could be a psychophysics example, clinical psych example, or neuroscience example, doesn't matter. I think it would simply help the reader better concretize the reasoning this way. This is entirely up to the authors, but I imagined that the broader audience that they (and I) want to reach would be more likely to engage if this was more explicitly related to their work.

[Editors' note: further revisions were suggested prior to acceptance, as described below.]

Thank you for submitting your article "Bayesian inference of population prevalence" for consideration by *eLife*. Your article has been reviewed by 2 peer reviewers, and the evaluation has been overseen by a Reviewing Editor and Timothy Behrens as the Senior Editor. The following individual involved in review of your submission has agreed to reveal their identity: Sam Ling (Reviewer #2).

Essential Revisions:

This is a paper that describes methods to move away from frequentist null hypothesis testing toward a Bayesian approach of estimate the prevalence of within-participant effects. This methodological advance should be widely applicable in many fields, and should promote a more robust evaluation of effects.

*Reviewer #1 (Recommendations for the authors):*

In their revised manuscript, Ince and colleagues have added a number of illustrative simulations and clarified several important points, substantially improving the final product. In particular, the simulated EEG analysis (new Figure 2) is an extremely clear and powerful demonstration of the advantages of prevalence estimation over population mean inference for a realistic problem. This is exactly the type of comparison that seemed to be missing in the initial submission, and its inclusion strengthens the paper substantially. The new simulations in Figures 4 and 6 also clarify the related points. Overall, I think the current manuscript is excellent, interesting, and timely.

Here I will respond to two of the authors' points specifically.

Authors: We avoid direct quantitative comparisons because the hypotheses tested (non-zero population mean, vs. posterior estimate of prevalence of within-participant effects) and statistical methods (frequentist vs. Bayesian) are different, although we do invite comparison of the different frequentist p-values (for null of mean zero vs global null of no participants with an effect) for the examples in Figure 1. We focused instead on qualitative arguments in favour of the prevalence view (better match to qualitative hypothesis of interest in psychology and neuroimaging, which we believe should better consider effects within individual behaving brains, and greater robustness given that the effect is shown independently in multiple participants).

This is spot on. The hypotheses tested by these methods are different, but the key question is how these (quantitative) hypotheses map to the implicit, qualitative hypotheses that are actually the subject of inquiry. On the discussion of this point, the current manuscript does not differ substantially from the original. Yet I still found myself much more convinced of its correctness this time around than the first. To this I credit the new simulations that the authors added (Figure 2 and 4). By directly comparing the results of the two methods, these simulations highlight how much better the hypotheses tested by prevalence estimation align with intuitive/qualitative hypotheses than do t-tests.

Minor aside: oddly, I think Figure 1 actually does the authors' arguments something of a disservice on this central point. Yes, it clearly illustrates how the two methods differ. But it presumes a world in which all the assumptions of the hierarchical population mean test are met. This leads to a situation where the standard method matches the reader's expectations (/qualitative hypotheses) much more closely than does the prevalence method. The authors acknowledge the weirdness of the two-sided effects seen in Figure 1B, and point out that this weirdness is actually implicit in the standard hierarchical model, but I worry that by this point the damage is already done. I'm not sure how best to do this – Figure 1 is a very clear illustration of the methods – but it could be worthwhile to consider reorganization that de-prioritizes this simple-but-weird example in favor of less weird examples like Figure 2.

Authors (regarding prevalence as a function of effect size): The main point of this section is to answer the potential criticism that we are dichotomising within participant results and to show that this is not a requirement for valid quantification of population prevalence. Prevalence as a function of effect size gives a fuller picture of the distribution of effect sizes observed. One could argue this is also given by a standard density plot, for example in a violin or raincloud plot, but these are descriptive properties of the specific data sample, without any formal inferential link to the population.

The new simulations shown in Figure 6 certainly clarify this point by showing examples with different underlying population statistics. I'm not specifically asking for anything more – I think this is sufficiently convincing as-is – but I think this would be stronger still with a more concrete example that shows how the prevalence/effect size curve could appear in different real-world situations, with implications for the reader's qualitative hypotheses. For an example like the question posed by Reviewer 2, this curve would look quite different if the individual subject standard deviation was not fixed, but was instead drawn from some distribution. Again, this is not critical, but a more real-world-like example would make this section more convincing.

*Reviewer #2 (Recommendations for the authors):*

The authors have done an admirable job addressing my questions. I'm eager for this work, which I hope to be influential in the field, to be out in the wild for others to adopt!

---

## [Author Response]

Summary:As you'll see below, the reviewers and I were all enthusiastic about the paper. That said, we all felt that a revision is required to provide concrete examples of when this method is preferred over frequentist approaches, as well as the general types of questions that this method is appropriate for addressing.

We thank the reviewers for their useful comments and apologise for the delay in producing our revisions. We have added additional simulations to compare directly with Null Hypothesis Statistical Testing (NHST) on the population mean and reordered the results presentation to emphasise the broad applicability of our Bayesian prevalence method.

In addition, we all felt that the focus of the paper should be expanded to give more attention to non-fMRI studies. For example, there is passing mention in the Discussion section about how this might apply to single unit studies, but that seems more of an afterthought at the moment. Thus, the paper is missing a chance to speak to, and influence, a much broader audience.

It was not our intention to focus on fMRI, although it’s possible that the citations reflected this area of research more than others. In fact, our approach has been developed primarily from behavioural and M/EEG studies. We revised the manuscript to clarify that the new approach is completely general with a simple abstract example to propose the concept. Also, we have now added simulations of typical event-related EEG experiments to motivate how Bayesian prevalence can be applied and how its results can differ from a second level t-test. The extension to single unit studies is a discussion point because we prefer to focus the examples to develop the critical point that human participants are the replication unit of interest in a broad range of psychology and neuroimaging studies. We have extended this discussion point with a more concrete example of how prevalence could be applied with neurons as the replication unit of interest.

Essential revisions:In addition to revising the text as indicated above (and in the more detailed reviews), it seems likely that some additional simulations may need to be run to address all of the points. This is particularly true of the suggestion to demonstrate a situation where prevalence estimation is more powerful than standard methods.

We want to emphasise that we don’t necessarily claim that Bayesian prevalence is more powerful than standard methods. In fact, the frequentist notion of power doesn’t directly apply to Bayesian estimation. Rather, we want to explain how Bayesian prevalence quantifies a different property of the sampled population, and thereby supports a conceptual shift of perspective, where the replication unit is the individual participant, and the proportion of participants showing a significant effect becomes the focus of statistical inference (in contrast to Null Hypothesis Significance Testing of the mean across participants). There are situations where this different perspective shows effects or differences that the population mean is not sensitive to, and we have added such example simulations as requested by the reviewers (new Figures 2, 4, 6).

Reviewer #1 (Recommendations for the authors):In this paper, Ince and colleagues describe a novel set of Bayesian methods for estimating and working with "population prevalence" of an effect, which they pose as an alternative to NHST on the mean effect in the population. In turns this paper was tantalizing, exciting, and frustrating. I very much want the method the authors propose to be useful, but I was left without a clear sense of (1) what specific questions ("qualitative hypotheses") this method is appropriate for answering, and (2) why (quantitatively) this method should be preferred over others. Specific comments and suggestions are enumerated below.1. Please show us how prevalence estimation can be more useful than classical approaches! The authors convincingly argued that estimating prevalence amounts to replicating an experiment in each individual subject, a shift in perspective that may help alleviate issues of non-replicability in psychology and neuroimaging (I am very sympathetic to these arguments!). However, it was somewhat frustrating that none of the computational experiments seemed to directly contrast prevalence estimation vs. population mean estimation. If prevalence is indeed a superior way to conceptualize effects in a population, then there should be some experiment where prevalence and population mean give different results for testing the same qualitative hypothesis. For example, in the discussion the authors bring up the issue of uninteresting heterogeneity across subjects, which can reduce the effectiveness of standard approaches (i.e. where the precise anatomical location or timing of an effect is variable across subjects). This paper would be much more convincing if it could experimentally demonstrate a situation like this (in simulation, of course), where prevalence estimation might truly be more powerful than standard methods.

In the original submission, we tried to strike a balance between a clear conceptual exposition and lists of practical examples, because the breadth of possible qualitative hypotheses to which Bayesian prevalence can be applied is so broad. In fact, pretty much any experimental question or test that can be considered at the per participant level can be framed as a population prevalence question, as an alternative to the typical population mean inference. So, in the revised manuscript, we keep the examples generic in the hope that readers could export the conceptual shift embedded within the prevalence concept to any questions that might arise from their own work.

We avoid direct quantitative comparisons because the hypotheses tested (non-zero population mean, vs. posterior estimate of prevalence of within-participant effects) and statistical methods (frequentist vs. Bayesian) are different, although we do invite comparison of the different frequentist p-values (for null of mean zero vs global null of no participants with an effect) for the examples in Figure 1. We focused instead on qualitative arguments in favour of the prevalence view (better match to qualitative hypothesis of interest in psychology and neuroimaging, which we believe should better consider effects within individual behaving brains, and greater robustness given that the effect is shown independently in multiple participants).

We would argue the simple generic simulation of the standard hierarchical population model shown in Figure 1 does demonstrate a situation where prevalence and population mean give different results for testing the same qualitative hypothesis (presence of a non-zero experimental effect in the population). This simulation directly contrasts inference of non-zero population mean (color of population curves, reported p-values on the left), with the prevalence approach (p-values for rejection of global null on the right, posterior prevalence curves in panel E). For example, the experiment simulated in Figure 1B shows very different results from the two approaches to testing the qualitative hypothesis “do participants in the population show a non-zero effect?”. The population mean provides no evidence for this (p=0.52), whereas the prevalence perspective provides very strong evidence (global null, p=2.2x10^-16^; purple posterior distribution in Panel 1E).

To give further intuition into situations where the NHST and Bayesian Prevalence can differ, we have added two new simulations of more concrete and realistic situations with similar divergence between the two approaches (Figure 2). These additional simulations don’t depend on symmetrical signed effects as the examples of Figure 1 (although we would like to emphasise that the presence of symmetrical signed effects is directly implied by the null hypothesis employed whenever a second-level t-test is used). The new Figure 2 shows simulated EEG data for two situations where prevalence can quantify robust population effects that are not visible from the population mean perspective. Panel A illustrates the alignment problem, robust effects are obtained in each participant, but at different post-stimulus times. Panel B illustrates the homogeneity problem, here the participants are heterogeneous, with 50% showing a strong effect and 50% showing no effect. In both cases the population mean NHST fails to show an effect when correcting for multiple comparisons over time. In contrast, the prevalence method does show strong evidence for effects at the population level (with the same multiple comparison correction applied within-participants).

As an aside, it was not our intention to suggest between participant heterogeneity is uninteresting; only that it can present a problem for traditional population-mean based approaches. In fact, we think exploring individual differences in anatomy and function is an important area of research. Our point is that Bayesian prevalence can provide an inferential link to the population which is currently missing in such cases.

2. Tell us more about what inferences can actually be made using prevalence estimation! The experiment in Figure 1 lays out the differences between prevalence and population mean estimation quite clearly, but the conclusions are much less clear. The authors "suggest that often, if the goal of the analysis is to infer the presence of a causal relationship within individuals in the population, the within-participant perspective may be more appropriate" – but more appropriate for what? Exploring-specifically at this point in the paper-which types of qualitative hypotheses (or some specific examples?) can be meaningfully addressed using prevalence analyses would, in my opinion, greatly strengthen the authors' case.

We aimed to present the prevalence approach as a completely general second level approach to drawing population level inferences from data obtained from multiple participants. In this sense, it is completely general and can test any qualitative hypothesis that can be quantified within individual participants (either via an effect size, or a within-participant NHST). Our argument is that, in many cases, the within-participant perspective may be more appropriate for quantifying the presence of a causal relationship (which could be any relationship tested with a randomised experiment and quantified with any test statistic or model). It can be applied to everything from behavioural modelling (including uncertainty, risk, expectation etc.), reaction time measures, multivariate decoding, encoding models for complex naturalistic stimuli, simple neuroimaging contrasts (face vs house), spectral analyses (phase or power response, phase-amplitude coupling etc.).

We have now moved the paragraph covering the breadth of possible applications to the start of this section to ensure it is in the front of the readers mind when considering the results of the simulations in Figure 1.

3. The argument that prevalence estimation is more sensitive to trials per participant than number of participants (Figure 2E) seems circular. The authors define prevalence as the fraction of subjects with "true positive (significant) test results" rather than the fraction where the null hypothesis is actually false, because they can only find a lower bound on the latter measure. Given this definition, the finding that "prevalence greater sensitivity to the number of trials per participant, and less sensitivity to the number of participants" (which is seemingly framed as the main conclusion of the analyses in Figure 2?) is highly unsurprising.

We agree this result is unsurprising when due consideration is given to the different perspective of prevalence. However, we still like to show it explicitly as most discussion of statistical power and power analysis in the psychology, neuroscience and neuroimaging literature considers only number of participants. So, we think it is important to emphasise this point – that the different perspectives have different properties, and that these give very different views in terms of experimental design. We also think these simulations can be used directly, or the scripts easily repurposed to provide simulation based “power analysis”. While not formal frequentist power analysis, these results could nevertheless provide some motivation for justifying number of participants in a particular study design. We have moved this to the end of the Results section to de-emphasise.

4. How does estimating group differences in prevalence compare to estimating differences in group means? The experiment shown in Figure 3 is tantalizing, but what I really wanted to know here is whether (and under what conditions) comparing groups using prevalence might be more powerful than comparing groups using traditional models.

They are quantifying fundamentally different properties of the populations. In Figure 4, we have added an additional simulation to illustrate the differences between t-test for difference in means and prevalence difference between two groups. We show two groups whose means do not significantly differ, but for which there is evidence for a clear difference in the proportion (of the population, not just the samples) that shows a statistically significant within-participant effect. If the population mean is the primary interest, then the t-test should be used. If the proportion of participants who show the effect is of interest, then Bayesian prevalence provides a valid methodology for making statistical claims at the population level. Crucially, these are valid quantitative statistical statements about the population, and not just a case study of the specific collected samples, which is the criticism traditionally levelled at within-participant statistics, e.g. “It should be acknowledged that this approach only allows for statements that pertain to the existence and magnitude of effects in those subjects, rather than in the populations those subjects are drawn from.” https://doi.org/10.1523/JNEUROSCI.0866-20.2020

5. The analysis of prevalence as a function of effect size is missing context and conclusions. Ultimately, I was left without any sense of the utility of estimating prevalence for different effect sizes after reading this section (and Figure 4). What is this specific procedure useful for? What kinds of hypotheses can it test? How does it compare to other methods? Answering these questions would undoubtedly strengthen the paper.

The main point of this section is to answer the potential criticism that we are dichotomising within participant results and to show that this is not a requirement for valid quantification of population prevalence. Prevalence as a function of effect size gives a fuller picture of the distribution of effect sizes observed. One could argue this is also given by a standard density plot, for example in a violin or raincloud plot, but these are descriptive properties of the specific data sample, without any formal inferential link to the population.

The prevalence as a function of effect size provides a model-free valid inferential link to the population and can be interpreted as Bayesian evidence for prevalence at the population level, rather than as a property only of the observed sample. Figure 6 shows six new simulations to illustrate situations where prevalence based on within-participant NHST at *p=0.05* is equivalent (panels A,B,C and D,E,F respectively), but the prevalence effect size plot gives further information, at the population level, that clearly distinguishes the simulated populations.

Reviewer #2 (Recommendations for the authors):This manuscript advocates for a shift in the statistical (and methodological) approaches taken in neuroscience and psychology, wherein effects are considered from the perspective of prevalence, rather than population-centric NHST. They provide simple yet compelling simulations to make their case, and introduce the bayesian prevalence method, along with code to implement on one's own. This manuscript is incredibly well written, the simulations are tightly constructed, and will, I hope, become widely read by the neuroscience and psychology community. It's a straightforward treatment of a statistic that the field needs to consider shifting towards, and the ability to generalize to any NHST will no doubt make this user friendly for most all researchers. I will readily admit that this was a method that my own lab has been in search of for a while, and we plan to deploy this in our own work. Below, I outline my questions/comments, but they are largely meant to support clarification to a broader audience.

We thank the reviewer for their comments.

1. While the authors promote a Bayesian treatment of the estimate of prevalence, they readily admit that a frequentist approach is suitable as well. It would be worth detailing a bit more the advantages and disadvantages of the two, if any. Particularly because the frequentist approach to neuroimaging has been out in the wild for a while now.

We have added some comments on this to the Discussion section.

2. The authors simulate individuals within the population as having equal within-subject (or unit) variability. To what degree are the assumptions for prevalence estimation upheld in the case where individual subjects have unequal variance?

The only assumptions for the Bayesian prevalence are that the participants are independent, and the false positive rate is properly controlled to the nominal α level within each participant. So long as the within-participant test is robust there is no problem with differences in variance between participants. We show here Author response image 1, a version of Figure 1 generated with between-subject standard deviation randomly sampled from a log-normal distribution with mean log(10) and standard deviation log(3). Prevalence decreases w.r.t. Figure 1 in the paper because some participants have much larger variance and so there are fewer within-participant significant results, but the prevalence estimation is still valid and the interpretations remain the same. We have added text making this point explicit in section “How to apply Bayesian prevalence in practise”.

**Author response image 1. sa2fig1:** Same as Figure 1 from manuscript, but within-participant standard deviation is sampled from a log-normal distribution with mean log(10) and standard deviation log(3).

3. I assume this rests on the assumption of independence between units of replication (ie subjects, voxels, ROIs, cells). Is this true? And how robust is this method to a lack of independence? I ask because if one were carrying this method out on a voxel-by-voxel basis, within a participant, there's quite a bit of non-independence between these voxels. And likely same goes for between adjacent ROIs, or even between time points. It would be helpful if the authors dedicated some text to this assumption.

The method does assume independence between “replication units” (most commonly thinking of participants) and so doesn’t address any hierarchical structure (e.g. time of day, time of year, testing lab, characteristics of university subject pool vs full population etc.). We would argue this is common to many statistical methods and needs to be evaluated by the particular scientific field as a “non-statistical” consideration (Eddington and Onghena, “Randomization Tests”, CRC Press). For example, the second-level t-test used for NHST of the population mean also assumes the participants are independent samples of the population.

We agree that treating voxels or ROIs as the replication unit of interest would make this problem particularly pronounced, and we haven’t suggested that as an intended application. We propose to do population prevalence across participants (ie. the proportion of the population that would be expected to show an effect in a particular voxel or region), which is different to treating voxels as the replication unit of interest and attempting to estimate the prevalence of an effect in a population of voxels (within a subject, the entire population of voxels for that individual is available). To simplify, throughout the paper we have participants as the replication unit of interest over which prevalence is calculated. The multiplicity which results from multiple time points, sensors or voxels is dealt with using existing multiple comparison correction techniques, but applied at the individual participant, rather than the population level. We hope the new simulations in Figure 2 make this more explicit (prevalence is calculated over participants, either for a time window, or for each time point, but prevalence is not carried out over time points).

Another possible replication unit which we discuss is to consider well isolated single unit neuron recordings. We have expanded this point in the discussion with a clearer example application. In this case, we propose combining neurons from multiple animals to determine the proportion of neurons in a particular region (where the population is all neurons across all animals). Again, the issue of whether it is valid to ignore the hierarchical structure is a problem-specific non-statistical consideration that the researcher must address. Perhaps it would be more reasonable to combine cells recorded in multiple sessions from the same animal (separate electrode insertions in the same brain region) to make an inference about the prevalence of an effect within the population of cells in that region in that individual if there is reason to suspect large variation between individual animals.

4. This comment is more to aid in promotion of the work, but I think the introduction and walkthrough of the simulation would be better serviced by using a case example throughout. It could be a psychophysics example, clinical psych example, or neuroscience example, doesn't matter. I think it would simply help the reader better concretize the reasoning this way. This is entirely up to the authors, but I imagined that the broader audience that they (and I) want to reach would be more likely to engage if this was more explicitly related to their work.

We have kept the simulations the same, but we have moved up to the start of this section the paragraph discussing the breadth and generality of possible applications, inviting readers to keep in mind the most relevant application to their work when considering the results of the simulations in Figure 1.

[Editors' note: further revisions were suggested prior to acceptance, as described below.]

Reviewer #1 (Recommendations for the authors):In their revised manuscript, Ince and colleagues have added a number of illustrative simulations and clarified several important points, substantially improving the final product. In particular, the simulated EEG analysis (new Figure 2) is an extremely clear and powerful demonstration of the advantages of prevalence estimation over population mean inference for a realistic problem. This is exactly the type of comparison that seemed to be missing in the initial submission, and its inclusion strengthens the paper substantially. The new simulations in Figures 4 and 6 also clarify the related points. Overall, I think the current manuscript is excellent, interesting, and timely.

We thank the reviewer for their comments. The paper evolved originally from a short perspective piece. We were aiming to keep the presentation general without locking in to one particular modality (like EEG) that not all readers might be familiar with. But we agree adding Figure 2 improves the presentation a lot and thank the reviewer for the suggestion.

This is spot on. The hypotheses tested by these methods are different, but the key question is how these (quantitative) hypotheses map to the implicit, qualitative hypotheses that are actually the subject of inquiry. On the discussion of this point, the current manuscript does not differ substantially from the original. Yet I still found myself much more convinced of its correctness this time around than the first. To this I credit the new simulations that the authors added (Figure 2 and 4). By directly comparing the results of the two methods, these simulations highlight how much better the hypotheses tested by prevalence estimation align with intuitive/qualitative hypotheses than do t-tests.

We thank the reviewer for such a frank opinion, which we of course share. However, we don’t want to fall into the trap of statisticians telling users what they want to know http://daniellakens.blogspot.com/2017/11/the-statisticians-fallacy.html We try to make the argument that the population mean is not the only population property we can formally quantify. In many cases prevalence might be of interest and particularly so in cognitive domains, where we would argue that prevalence might often better match the qualitative scientific question of interest. We agree that the new Figures make this point much more clearly and hope our new Figure 6 will similarly add to the weight of the argument. We thank the reviewer for their input that helped us understand how to better communicate the shift of perspective from means to prevalence.

Minor aside: oddly, I think Figure 1 actually does the authors' arguments something of a disservice on this central point. Yes, it clearly illustrates how the two methods differ. But it presumes a world in which all the assumptions of the hierarchical population mean test are met. This leads to a situation where the standard method matches the reader's expectations (/qualitative hypotheses) much more closely than does the prevalence method. The authors acknowledge the weirdness of the two-sided effects seen in Figure 1B, and point out that this weirdness is actually implicit in the standard hierarchical model, but I worry that by this point the damage is already done. I'm not sure how best to do this – Figure 1 is a very clear illustration of the methods – but it could be worthwhile to consider reorganization that de-prioritizes this simple-but-weird example in favor of less weird examples like Figure 2.

We understand what the Reviewer is suggesting, weighted both options, even tried to change the presentation as suggested. However, after careful thinking the logic of presentation of current Figure 2 still requires that the prevalence method is clearly defined. The best way to do this is to introduce it step by step (from the idea of within-participant significance providing evidence against a population effect given by the global null). So, in the end, we decided to leave current Figures 1 and 2 figures in this order, for the purpose of logical clarity of presentation. We have added a sentence to signpost Figure 2 in the last paragraph of the *Population vs within-participant statistical tests* section.

Authors (regarding prevalence as a function of effect size): The main point of this section is to answer the potential criticism that we are dichotomising within participant results and to show that this is not a requirement for valid quantification of population prevalence. Prevalence as a function of effect size gives a fuller picture of the distribution of effect sizes observed. One could argue this is also given by a standard density plot, for example in a violin or raincloud plot, but these are descriptive properties of the specific data sample, without any formal inferential link to the population.The new simulations shown in Figure 6 certainly clarify this point by showing examples with different underlying population statistics. I'm not specifically asking for anything more – I think this is sufficiently convincing as-is – but I think this would be stronger still with a more concrete example that shows how the prevalence/effect size curve could appear in different real-world situations, with implications for the reader's qualitative hypotheses. For an example like the question posed by Reviewer 2, this curve would look quite different if the individual subject standard deviation was not fixed, but was instead drawn from some distribution. Again, this is not critical, but a more real-world-like example would make this section more convincing.

We have produced a new Figure 6, which uses the same EEG simulation setting as Figure 2 to motivate three different situations where the overall prevalence of *p=0.05* is similar, but the effect size prevalence curve reveals interpretable differences between the simulated data sets. We hope this makes a more intuitive and clearer link between data and effect size prevalence results. As these three new examples cover the main features of the old Figure 6 (different variances and subgroups) we replaced the old figure with this new one.